# Exceptionally high work density of a ferroelectric dynamic organic crystal around room temperature

Durga Prasad Karothu [1], Rodrigo Ferreira[1], Ghada Dushaq[2], Ejaz Ahmed [1], Luca Catalano [1], Jad Mahmoud Halabi [1], Zainab Alhaddad[1], Ibrahim Tahir [1], Liang Li [1,3], Sharmarke Mohamed [4], Mahmoud Rasras[2] & Panče Naumov [1,5] ✉

Dynamic organic crystals are rapidly gaining traction as a new class of smart materials for energy conversion, however, they are only capable of very small strokes (<12%) and most of them operate through energetically cost-prohibitive processes at high temperatures. We report on the exceptional performance of an organic actuating material with exceedingly large stroke that can reversibly convert energy into work around room temperature. When transitioning at 295–305 K on heating and at 265–275 K on cooling the ferroelectric crystals of guanidinium nitrate exert a linear stroke of 51%, the highest value observed with a reversible operation of an organic single crystal actuator. Their maximum force density is higher than electric cylinders, ceramic piezoactuators, and electrostatic actuators, and their work capacity is close to that of thermal actuators. This work demonstrates the hitherto untapped potential of ionic organic crystals for applications such as light-weight capacitors, dielectrics, ferroelectric tunnel junctions, and thermistors.

[1] Smart Materials Lab, New York University Abu Dhabi, PO Box 129188 Abu Dhabi, UAE. [2] Division of Engineering, New York University Abu Dhabi, PO Box 129188 Abu Dhabi, UAE. [3] Science and Engineering Department, Sorbonne University Abu Dhabi, PO Box 38044 Abu Dhabi, UAE. [4] Department of Chemistry, Green Chemistry & Materials Modelling Laboratory, Khalifa University of Science and Technology, PO Box 127788 Abu Dhabi, UAE. [5] Molecular Design Institute, Department of Chemistry, New York University, 100 Washington Square East, New York, NY 10003, USA. ✉email: pance.naumov@nyu.edu

Molecular crystals comprise an emerging class of materials that contribute to the engineering materials' toolbox structures that are endowed with mechanical softness, long-range structural order, and anisotropy in their physical properties[1–4]. At much lower density than many other engineering materials, the weak intermolecular interactions in organic crystals can effectively absorb elastic energy that had accumulated as a result of the development of mechanical strain, thereby effectively expanding their plastic regime. In certain applications, this plasticity of molecular crystals could be utilized to compensate for the common brittleness of some inorganic materials such as the technical ceramics. Organic crystals have elastic moduli that are intermediate between those of biogenic and inorganic materials and long-range order, which is atypical for mesophases at densities comparable to those of soft biological tissues[5]. These assets, that are rooted in the low-energy, soft, and directional intermolecular interactions in their structures, occasionally come with dynamic, self-healing, and restorative capabilities and an ability of diffusion of their molecules across free surfaces and inter-particle interfaces[6–13]. Yet, perhaps the most valuable asset of using dynamic organic crystals is not considered to be their response time, but rather their nature as lightweight materials, particularly in applications that require minimal weight per volume of the material such as in medical devices, prosthetics, compliant electronics, and soft robotics. However, although there are no available quantitative and encompassing systematic studies that would estimate the range of energies that dynamic molecular crystals can deliver in the form of work, presently they do not appear to be competitive with other, more robust materials such as polymers[14]. Two challenges with real-world applications of organic crystals (for example, in microfluidics) have been their slow mechanical response and the small strokes they are able to exert[15]. Much of the research of late has focused on simple deformation of molecular crystals induced by excitation with light, such as photoinduced bending of organic crystals[16,17]. Although in some cases the change in crystal shape is reversible, and the crystal can be deflected many times, the deformation and shape restoration occur on a timescale of seconds to minutes—an unacceptably slow response in view of applications, which would require structure switching on a scale of milliseconds or faster[18]. The slow rate of crystal bending is usually inherent to the small light absorption cross-sections that translates into low conversion yield, and the weak coupling between the photochemical transformation and the ensuing mechanical deformation. Using slender crystals to increase the yield comes at the cost of proneness to fracture, which becomes impractical in dense, fluid, and/or turbulent environments[6].

A much more viable approach to making organic crystals competitive with other materials would require very fast mechanical reconfiguration, such as those that accompany some crystal-to-crystal phase transitions. Cooperative phase transitions, and specifically martensitic transitions, are known to be particularly fast transformations[12,19–21]. Indeed, several recent examples of mechanical effects based on phase transitions in organic crystals, inorganic crystals, hybrid crystals, and a few ionic compounds (both inorganic and organic) have indicated the hidden potential of these materials for fast actuation[22–45]. Since the molecular movements are hardly controlled once they are initiated, the realization of practical applications by using dynamic crystals poses challenges with the collective molecular movement and the related lack of control over the mechanical response. With regards to the shape transformation, some of the reported dynamic crystals are prone to disintegration, deterioration, splintering, and deformation during the phase transition, which hampers significantly the implementation of these materials into devices with cyclic operation[31–37]. However, recent reports on alteration of the thermal hysteresis of phase transitions by manipulating the phase boundaries[46] and the influence of second-order phase transitions on molecular configuration in a nickel complex have demonstrated the possible controllability of dynamic effects[47]. While each of these reported materials sets a milestone in the quest for an applicable dynamic organic crystal, the typical strokes that can be obtained by their expansion or contraction rarely exceed several percent, and are typically less than 12%. Moreover, the phase transition in some of the materials is not reversible and thus the material cannot be cycled. In many cases, the phase transition occurs at a temperature much higher than the ambient temperature, and thus the general application of such actuators would be energetically prohibitive. In turn, amplification of molecular-scale motions to macroscopic scale by organic crystals, which would make these materials attractive for engineering design, has not been accomplished yet. Here, we report that single crystals of guanidinium nitrate (GN), a ferroelectric material that undergoes rapid and reversible first-order phase transition around room temperature, exhibits the highest reported stroke upon transitioning between two phases due to collective reorientation of its ions in the crystal lattice. During our broader screening for materials with thermosalient (sudden motion or shattering induced by a phase transition), actuating, and other dynamic properties we have synthesized and analyzed a number of guanidine derivatives. While most of these derivatives did not show thermosalience or other dynamic behavior, a few were found to exhibit interesting properties. The material reported here was particularly interesting for its extraordinary expansion upon heating. The mechanism of the phase transition was investigated using dispersion-corrected density functional theory (DFT-D) methods, which revealed that the rapid martensitic phase transition in GN is a consequence of the energetically favored collective rearrangement of hydrogen-bonded ions. The potentials for application of single crystals of this material as an electronic material with ferroelectric properties are demonstrated by dielectric, capacitance, conductance, and current measurements.

## Results and discussion

**Characterization of the phase transition.** Single crystals of GN were grown by slow evaporation from a saturated aqueous solution below 285 K at ambient pressure following a reported procedure[48] (the details for preparation are provided in the "Methods" section). Below 285 K single crystals of the low-temperature polymorph (form I) were obtained as colorless elongated hexagonal needles, plates, or prisms with well-developed faces (Fig. 1a; Supplementary Fig. 1). Conducting the crystallization above 296 K resulted in a polycrystalline material. Differential scanning calorimetry (DSC) confirmed that crystals of form I undergo endothermic phase transition to form II at 295–305 K on heating, and form II reverts back to form I on cooling by an exothermic transition at 275–265 K (Fig. 1d). The sharp transition, relatively large thermal hysteresis of about 20 K, and distinct peak shape confirm that the phase transition between forms I and II is of first order. Single crystals or lightly ground crystals can be cycled between the two forms by repeated heating and cooling with a small shift in the transition temperature between the first and the second thermal cycle (Supplementary Fig. 2). Variation in the transition of GN crystals is observed during heating and cooling cycles with respect to the change of heating/cooling rates (Supplementary Fig. 3). In single crystals, the phase transition can be observed with a naked eye during heating and cooling as a drastic change in the crystal size (Supplementary Movies 1 and 2). When transitioning to form II, needle-shaped crystals of form I expand strongly and visibly

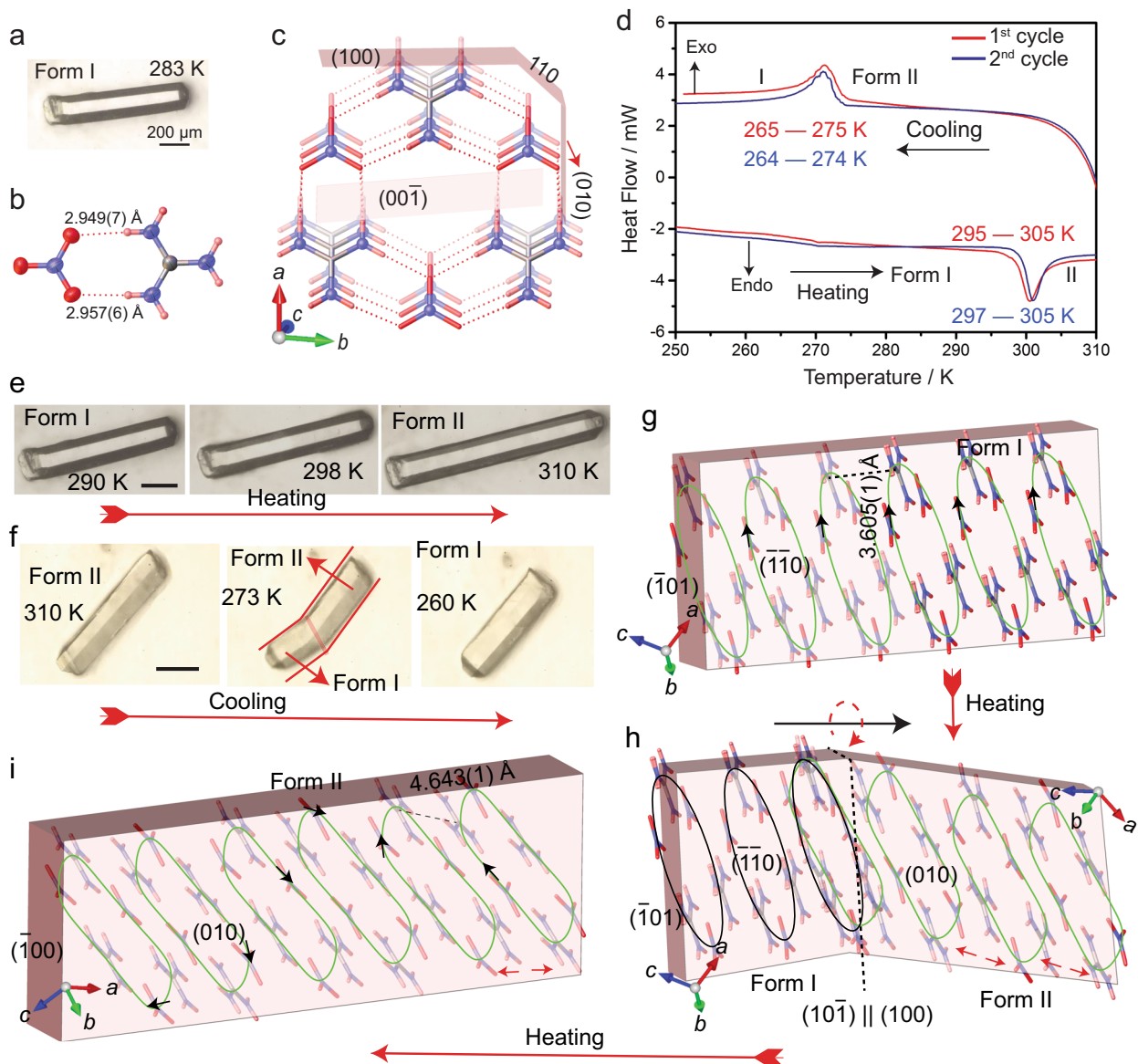

**Fig. 1 Crystal habit and structure, and phase transition between forms I and II of GN. a** Typical prismatic crystal habit of form I obtained by crystallization from solution. **b** ORTEP diagram of the main hydrogen bonding motif in the crystal structure at 250 K. The thermal ellipsoids of the non-hydrogen atoms are shown at the 50% probability level. **c** Crystal structure of form I viewed in the [001] direction. The hydrogen bonds are shown as dotted lines. **d** Thermal effects accompanying the transition between forms I and II recorded by DSC (two thermal cycles are shown). Note the offset in the maximum transition temperature between the two cycles. **e, f** Changes in size and shape of single crystals of GN taken twice over the phase transition. Crystals converted from form I to form II by heating (**e**) and from form II to form I by cooling (**f**) between 260 and 310 K are shown. Note that the crystal does not show any visible deterioration. **g, h, i** Crystal packing comparison of the form I (**g**), partially converted crystal (**h**) form II (**i**) and analyzed by single crystal X-ray diffraction. Scale bars: **e** 300 μm, **f** 200 μm.

along their longest crystal axis, while they shrink across their width (Fig. 1e). Measurement of the crystal size before and after the transition showed that the crystals expand by an impressive 51% during heating. Upon cooling, they shrink to their original size without any visible deterioration (Fig. 1f). Both transitions depend on the rate of heating, and the phase boundaries were observed at different heating rates during the phase transitions (Supplementary Fig. 4 and Supplementary Movie 3).

**Structural characterization.** Crystal structure analysis of forms I and II was performed by X-ray diffraction from crystals at 250 and 310 K, respectively (Supplementary Table 1). Form I crystallizes in the monoclinic space group $Cm$ ($a = 10.925(3)$ Å,

$b = 7.272(1)$ Å, $c = 3.605(8)$ Å and $\beta = 93.613(8)°$). The asymmetric unit cell consists of half formula unit with one guanidinium and one nitrate ion. The layered crystal structure is composed of ionic sheets parallel to the (001) plane consisting of alternating guanidinium and nitrate ions in hexagonal pattern (Fig. 1b, c; Supplementary Fig 5). The adjacent guanidinium and nitrate ions are bonded to each other through a couple of hydrogen bonds, with N···O distances of 2.957(6) Å and 2.949(7) Å. Each guanidinium cation is bonded to three nitrate anions through six hydrogen bonds, and each nitrate anion is bonded to three guanidinium cations through six hydrogen bonds. In form I, the ionic sheets are stacked atop each other at a distance of 3.605(1) Å. The 12-membered hydrogen-bonded rings of adjacent sheets are offset at a slippage angle of 76.8°, and the ions are

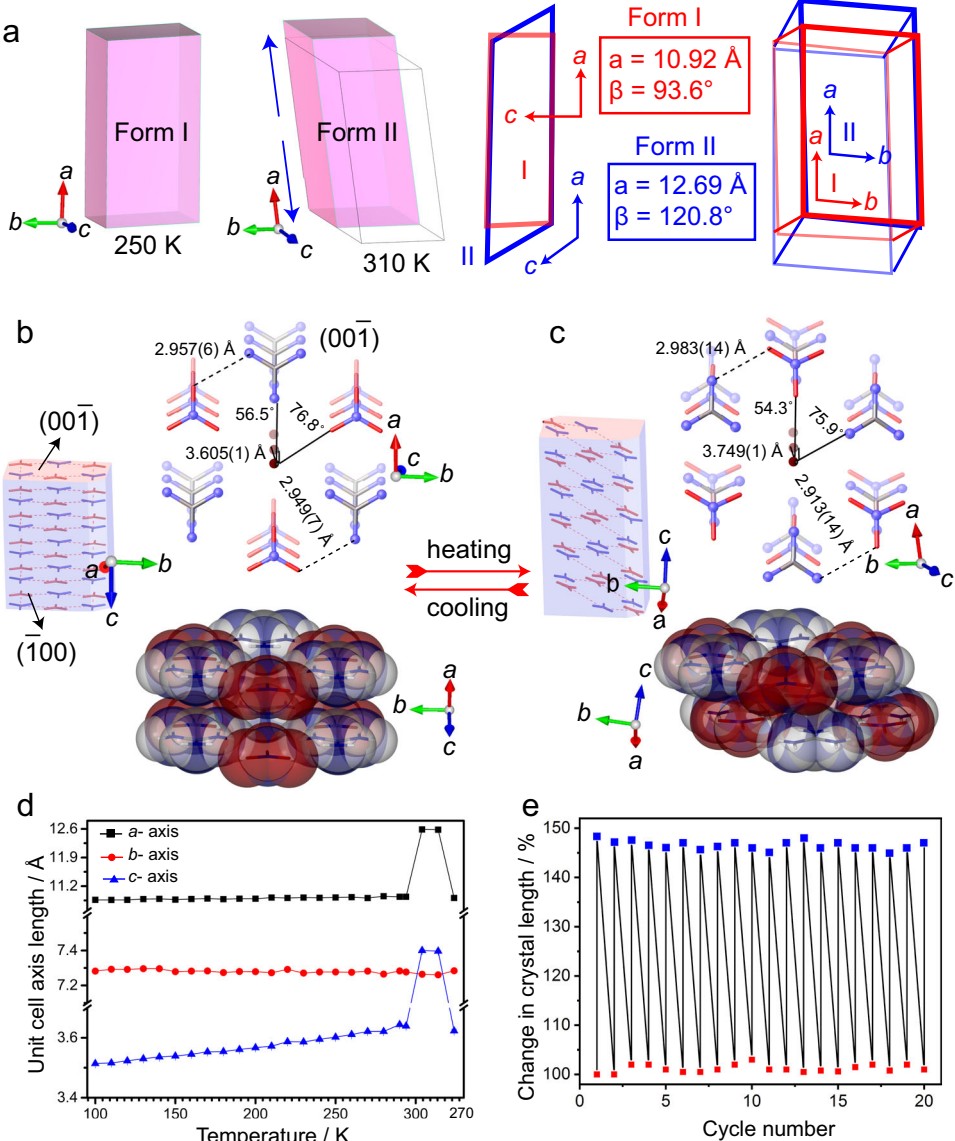

**Fig. 2 Changes in the crystal structure and crystal size and shape during the phase transition. a** Comparison of the unit cells before and after the phase transition. The values of the characteristic unit cell parameters in the two phases are given. The doubling of the $c$ axis that occurs during the transition from form I to II is not shown here. **b, c** Cartooned crystal habits of form I (**b**; 250 K) and form II (**c**; 310 K) shown with molecular orientation in form I (**b**) and form II (**c**) crystals viewed in the [001] and [100] directions. Upon heating from 290 to 310 K, the distance between the adjacent sheets containing nitrate and guanidinum ions increases from 3.605 to 3.749 Å while the adjacent hexameric layers are apparently twisted in respect to each other. **d** Changes in the unit cell parameters of form I before and during the reversible phase transition. **e** Cyclability of a good-quality single crystal over the phase transition monitored by the change in its length. The expansion and shrinkage of the crystal were recorded over 20 thermal cycles.

displaced along the $a$ axis (Figs. 1g and 2a, b). Upon transformation to form II, the hexagonal hydrogen-bonding pattern within the ionic sheets is preserved, while the arrangement of the sheets adjacent to each other changes to an antiparallel pattern (Figs. 1h, i and 2a, c). The crystal structure of form II is also monoclinic ($C2$, Supplementary Table 1; $a = 12.69(3)$ Å, $b = 7.283(2)$ Å, $c = 7.256(2)$ Å and $\beta = 120.828(9)°$). The sheets are placed atop of each other at a distance of 3.749(1) Å. The adjacent sheets are slightly offset with respect to each other, at a slippage angle of 75.9°. The guanidinium cations in the 12-membered ring and the overlapping guanidinium cations of adjacent sheets are rotated by about 60°. The positive and negative ions of two adjacent layers are placed in opposite directions, and are alternating in the (001) plane (Supplementary Fig. 6).

To investigate the change of the structure upon heating before the phase transition, the temperature-dependent variation in the unit cell parameters was recorded (Fig. 2d; Supplementary Table 2). The expansivities along the principal axes of form I prior to the phase transition and the respective expansivity indicatrix are available from Supplementary Fig. 7. The thermal expansion of the crystal of form I preceding the transition to form II is strongly anisotropic. The minor (X1), medium (X2), and major (X3) principal axes in form I are oriented along the [102], [010], and [001] directions, respectively. The crystal undergoes small biaxial negative expansion, with coefficients of $-4.98 \times 10^{-6}$ and $-6.81 \times 10^{-6}$ K$^{-1}$ parallel to $a$ and $b$, respectively, and a large ('colossal') uniaxial positive thermal expansion (PTE) parallel to $c$ with coefficient of $211.46 \times 10^{-6}$ K$^{-1}$. The volumetric thermal expansion (VTE) is $210.79 \times 10^{-6}$ K$^{-1}$. The positive axial and

VTE coefficients of form I are comparable to or exceed those reported for other compounds (Supplementary Table 3)[12,49–53]. The large uniaxial expansion along $c$ results in accumulation of sizeable strain, until the structure switches to form II. The transition is accompanied by abrupt changes of the unit cell above 295 K on heating and below 275 K on cooling. The unit cell variation confirms that the transition is reversible with a thermal hysteresis. Both polymorphs were found to be relatively soft[54–56] and susceptible to deformation under the effect of external pressure. The elastic modulus ($E$) and hardness ($H$) of form I are $E = 4.2 \pm 0.4$ GPa; $H = 0.1 \pm 0.02$ GPa on the (100) face whilst for form II they are $E = 3.5 \pm 0.2$ GPa and $H = 0.1 \pm 0.01$ GPa on the (100) face (Supplementary Fig. 9). DFT-D simulations reveal that the stacking of the ions in forms I and II is equienergetic. This favors the facile rearrangement of the hydrogen bonded sheets (Fig. 2b, c) with temperature, a process that is associated with a net release of energy upon rotation of the hydrogen-bonded sheets during the thermally induced phase transition (Supplementary Fig. 11a, b). The DFT-D simulations also reveal that the strain energy required to deform the cell can be as high as 11 kJ mol$^{-1}$ (Supplementary Fig. 11c, d). However, this is more than compensated for by the energy released upon re-arrangement of the hydrogen bonded sheets, which was found to be as large as $-15$ kJ mol$^{-1}$ during the phase transition (Supplementary Fig. 11a).

**Actuating performance**. Supplementary Movies 1 and 2 show the change in crystal size during the phase transition. As mentioned above, the single crystals strongly expand and contract by about 51% along their longest axis (parallel to the $ac$ plane), and this deformation is completely reversible. We also note that some of the longer crystals appeared to slightly bend during heating, in a motion that resembles snake slithering (Supplementary Movie 4 and Supplementary Fig. 10). This macroscopic mechanical effect could be caused by a mechanical instability. The forward phase transition is very fast: at a heating rate of 10 K min$^{-1}$, the transition of form I to form II occurs in less than a few milliseconds. At the same cooling rate, the transition of form II to form I takes more than a few seconds. Figure 3c shows graphically, and Supplementary Table 4 shows more quantitatively a comparison with the expansion of other dynamic crystals whose relative expansions have been reported in the literature. As it can be inferred from there, the elongation of GN is the second largest of all the reported materials, with the largest deformation observed in a CuQ$_2$–TCNQ complex[57]. However, while during the transition of CuQ$_2$–TCNQ the crystals shatter after the first expansion, in the case of GN the crystals retain their integrity and remain intact even after multiple cycles of switching between the two phases. Indeed, the single crystals of GN are mechanically robust and as shown in Fig. 2e, they do not show evidence of fatigue even after 20 cycles, a result that reflects their extraordinary resistance to fatigue. This robustness is not only a property that sets GN apart from the other martensitic crystals, including thermosalient crystals, which tend to disintegrate[1,6,7,12,31–37], but it also stands as one of the most essential prerequisites for practical applications in devices such as multiple-stroke actuators.

In order to assess the performance of this material for actuation, the forces generated by single crystals before and during the phase transition were directly measured. Each crystal was sandwiched perpendicularly to its axis of expansion between a rigid wall and the tip of a sensitive force sensor (Fig. 3a). The force by which the crystal pushed the sensor measured from six crystals of different sizes was 1.34–49 mN (Fig. 3b and Supplementary Table 5). The strokes measured from the six crystals were in the range 0.35–0.95 mm, since the stroke largely depends on the actual dimensions of the crystal. Figure 3c shows that crystals of GN generate the strongest force among the thermosalient crystals. Their force output is considerably higher than those of MEMS and electroactive polymers, and is comparable to those of polymer gels, some piezoelectric actuators, and nanomuscles. GN has an elasticity and hardness larger than most polymers, gels and elastomers, which explains its ability to generate larger force output without succumbing to the deformation that is typically observed with softer materials. As shown with the materials property plot in Fig. 3d, given their small size, the maximum force density (maximum force generated per unit volume) of GN crystals is indeed higher than many actuators including electric cylinders, ceramic piezoactuators, and electrostatic actuators.

The maximum work output of an actuator is defined as $W_{max} = F_{max} \cdot \Delta L$ where $F_{max}$ is the maximum force output and $\Delta L$ is the induced displacement or the stroke. A freely moving actuator (without load) can exhibit large displacement without any force generation, while a heavily loaded actuator reaches its capacity of maximum force generation with a zero displacement. In both cases, there is no work output. Therefore, the work output is an important metric since it considers both the displacement and the force generated in a single actuation response together rather than considering the two properties separately. Notably, the crystals of GN have a remarkable work capacity (work output per unit volume) and rank highly compared to other actuators (Fig. 3e). In this aspect, they are particularly close in performance to thermal actuators. To select an appropriate material for a specific task, important attributes (for example, force output and work capacity) of different actuators are usually compared and those that are within one order of magnitude apart are considered viable competitors for the function at hand.

**Electrical properties**. The internal electric dipoles of guanidinium nitrate, a ferroelectric crystal, are physically tied to its lattice, and thus anything that changes the physical lattice will also affect the strength of the dipoles and result in variation of the electrical characteristics. The electrical properties accompanied by the phase transition were investigated by $C$–$V$, $G$–$V$, and $I$–$V$ measurements. The $C$–$V$ curves provide direct information about the electric field inside the material, and are not directly related to the variation of carriers in the crystal. Figure 4c shows the $C$–$V$ curves at 200 kHz for forms I and II. As can be seen, the form II crystal shows seven-fold higher capacitance compared to form I. Similarly, form II exhibits 35-fold increase in conductance compared to form I at the same frequency (Fig. 4d). The $C$–$V$ and $G$–$V$ characteristics for both phases at different frequency sweep are summarized in Supplementary Table 6. The four types of polarization are electronic, ionic, dipolar, and interfacial, and all of them can effectively add to the capacitance and conductance values at low-frequency range. At high frequencies, the interfacial, dipolar, and ionic polarization contributions become insignificant, leaving only the contribution from the electronic polarization relevant. This frequency dependence can be clearly seen in both phases of the crystal. As shown in Supplementary Table 6, the capacitance/conductance decreases as the frequency increases from 1 to 200 kHz. This is due to the fact that at higher frequencies the temporal response of the interfacial dipoles is reduced, leading to a lack of rearrangement in the alternating field's direction. $C$–$V$ measurements at frequencies above 200 kHz (500 kHz–1 MHz) showed a better response of form I compared to form II, indicating a stronger carrier response.

The dielectric constants of both crystal phases were calculated from the values of the capacitance and conductance for the

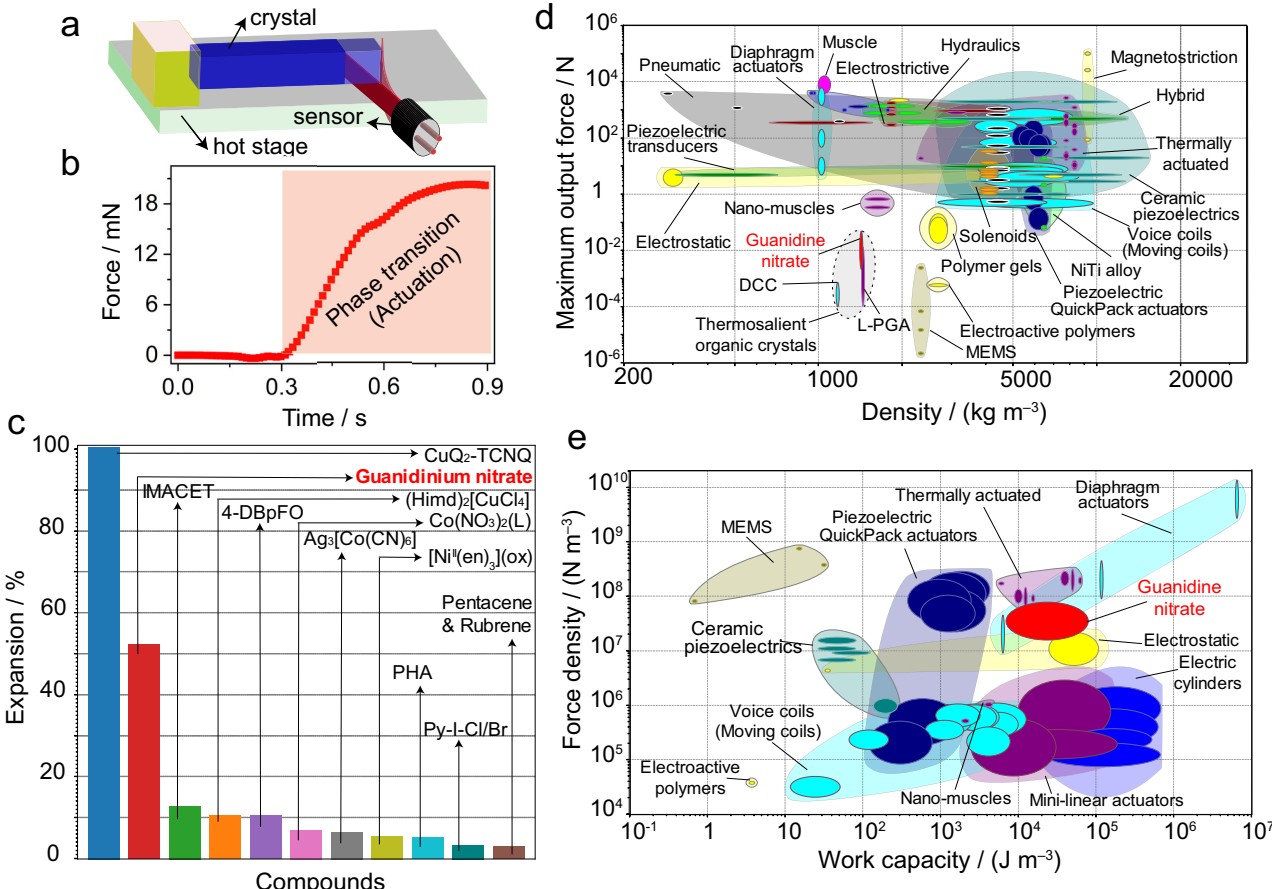

**Fig. 3 Evaluating the actuating performance of guanidinium nitrate crystals. a** A setup used to measure the force generated by GN single crystals upon thermal expansion. **b** Force measured during phase transition of form I to form II during heating as a function of time (in seconds). **c** Comparison of thermal expansion of GN to other organic crystals. **d** Maximum output force versus density of GN and two other thermosalient materials (L-pyroglutamic acid and a DABCO-carbazole cocrystal)[5,21,32] co-plotted together with other common actuator classes. **e** Work capacity versus density of GN co-plotted with the same attributes for other main actuator classes. The opaque bubbles represent the range of performance indices values of a particular material, while the other translucent envelopes group together materials belonging to the same actuator class.

Ag/crystal/copper structure at 100 kHz. The complex dielectric can be expressed as:

$$\varepsilon_1 = \frac{Y^*}{j\omega C_0} = \frac{C}{C_0} - i\frac{G}{\omega C_0} \qquad (1)$$

Here, $C$ is the capacitance, $G$ is the conductance, $Y^*$ is the admittance, and $\omega$ is the angular frequency of the electric field, calculated as $\omega = 2\pi f$. The dielectric constant ($\varepsilon_1$), at 100 kHz can be obtained from capacitance at voltage of 1 V.

$$\varepsilon = \frac{C}{C_0} = \frac{C d_{OX}}{A \varepsilon_0} \qquad (2)$$

where $d$ is the crystal thickness, $A$ is the plates area and $\varepsilon_0$ is the permittivity of air. The difference in the calculated value of dielectric permittivity between the low- and high-temperature phases amounts to about 30%. The onset of the increase in dielectric constant in the high-temperature region is related to the increase in the electric conductivity of the sample.

The fact that the internal electric dipoles can be forced to change their direction by application of an external voltage gives rise to hysteresis in this class of crystals (spontaneous polarization). The $C–V$ hysteresis is obtained by sweeping the gate voltage from −1.5 to +1.5 V and then immediately sweeping back to −1.5 V at 200 kHz. The form I crystal phase has a hysteresis width of ~900 mV, whilst a reduced width of ~500 mV was observed in the form II crystal (Fig. 4e, f). This can be explained

by the fact that high temperature disturbs the ferroelectric alignment and produces molecules that are free to reorient in an external field. Additionally, the quality of the interface between the silver and the crystal can contribute to the observed hysteresis, most probably by physical adsorption of a thin film of water on the crystal surface which can occur with testing in air.

The $I–V$ characteristic of Ag/form I-crystal/Ag and Ag/form II-crystal/Ag (metal-semiconductor-metal configuration) are shown in Fig. 5a. Both crystal phases demonstrate ohmic contact with silver (symmetric $I–V$ curves). The current density ($J$ = current/area) calculated for the two crystal phases was $J = 0.28$ mA cm$^{-2}$ and 0.8 μA cm$^{-2}$ for form I and form II, respectively. It is worth pointing out that during the transition from form I to form II, the crystal defects are the source of inner local stress and thus they can be responsible for the reduction seen in current density of form II (Fig. 5b). This is also consistent with the higher conductance values obtained for the form II crystal, where the defects can act as fixed charges that add up to the conductance values.

In summary, here we report and systematically rationalize the origin of the extraordinarily large thermal actuation in guanidinium nitrate crystals by comparing the crystal structures before and after the thermal phase transition. The impressive crystal actuation of this material originates from a first-order phase transition that is preceded by a colossal expansion parallel to the $ac$ plane. The main structural difference between the two forms is

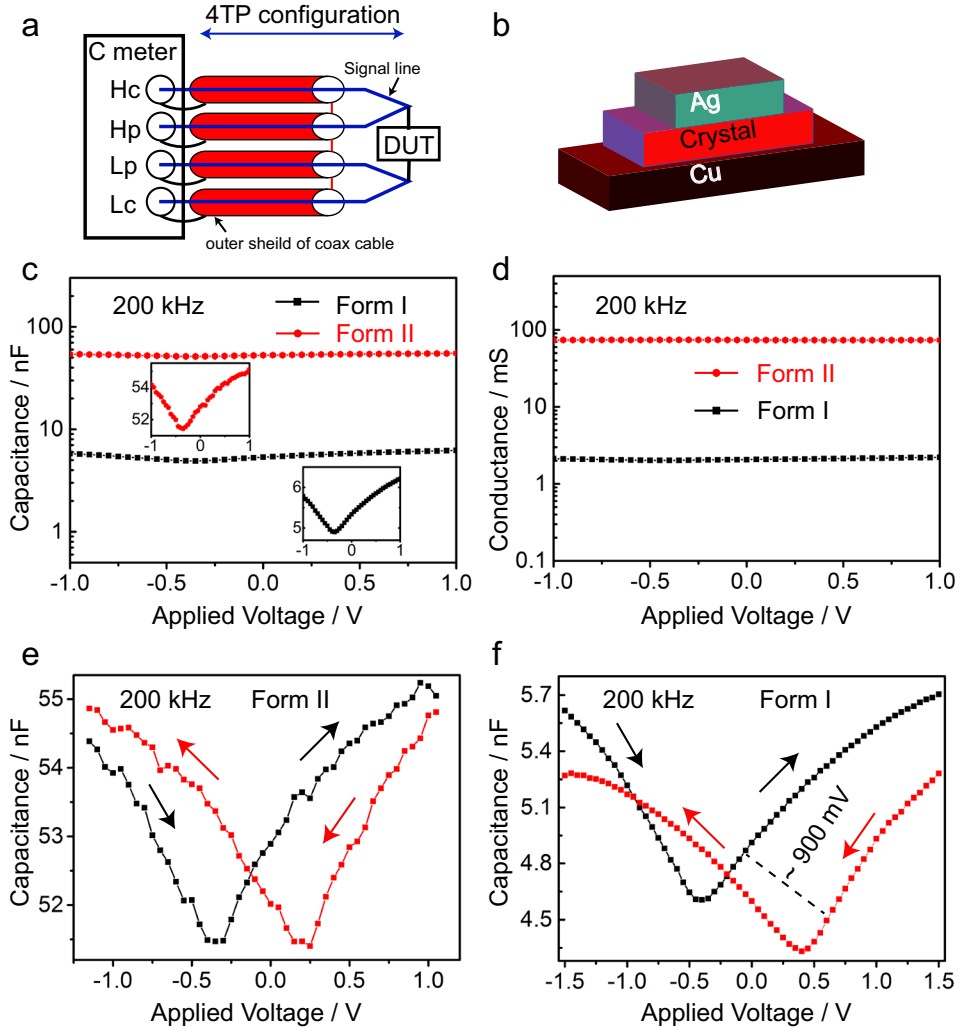

**Fig. 4 Electrical measurements of GN showing the frequency-dependent capacitance–voltage (*C–V*) and conductance–voltage (*G–V*) plots. a** Four-terminal-pair configuration (HC: high current, HP: high potential, LP: low potential, LC: low current). **b** Schematic of the fabricated Ag/crystal/Cu capacitor. **c** Capacitance–voltage (*C–V*) measurement on a single crystal of form I GN. Later the same crystal was heated, transformed to form II, and used for the *C–V* measurements. **d** Conductance–voltage (*G–V*) measurements of form I and form II. **e, f** The *C–V* hysteresis.

that in form I the sheets containing guanidinium and nitrate ions have parallel orientations; during phase transformation to form II these sheets rearrange to antiparallel orientations with significant changes in the intermolecular interactions, and also the distance between adjacent sheets. DFT-D simulations reveal that the ion stacking in the two phases is equienergetic, favoring rapid switching between them. The calculations also show that the energy penalty associated with the cell deformation can be compensated for by the energy released upon rotation of the ions during the thermally induced phase transition. The crystal of form I is actuated by expansion parallel to the *ac* plane with excellent reversibility for more than 20 cycles of the thermally induced structural phase transition without visible deterioration. The measured changes in intermolecular interactions in forms I and II are significantly affected by the electrical performance of these crystals. Notably, the difference in the calculated values of the dielectric permittivity between forms I and II amounts to about 30%. Furthermore, the current density measurements for both crystal forms show variation in their carrier response. Interestingly, the frequency dependence of the *C–V* and *G–V* characteristics for both crystal phases are very similar to inorganic semiconductors where they exhibit a decrease in capacitance/conductance as the frequency is increased from 1

to 200 kHz. In an attempt to verify the generality of the results, we have also synthesized and analyzed a number of other derivatives of guanidine. While most of these derivatives did not show thermosalience or other dynamic behavior, a few of them did exhibit interesting dynamic properties. The material reported here was particularly interesting for its extraordinary expansion. We believe that designing materials with desired dynamic properties is a far-reaching goal and is likely to require a lot of systematic effort to be realized in the future.

## Methods

**Materials**. Guanidinium nitrate (GN) was obtained from Sigma Aldrich and was used for crystallization without further purification. The solvents were obtained from commercial suppliers and were used without purification.

**Crystallization**. Crystallization was based on slow evaporation of an acetone/water mixture. First, 200–400 mg of GN was taken into a conical flask. Approximately 5–10 mL of acetone was added and the conical flask was put inside the sonicator for 20–30 min at room temperature. Then, small amounts of water were added to the conical, until all the solid was dissolved. The solution was then filtered twice and the filtrate was collected in a crystallization dish, which was placed on top of ice to prevent crystallization at room temperature. Finally, the crystallization dish was taken to a crystallization room with temperature lower than 286 K at all times. GN crystals were observed in 3–4 days.

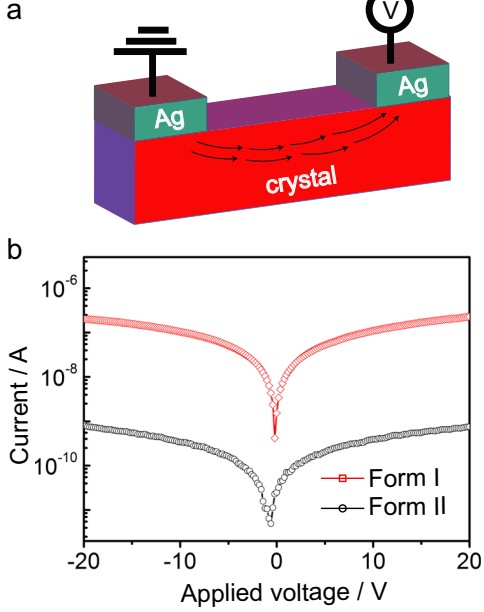

**Fig. 5 The I–V (current–voltage) characteristics of form I and form II GN crystals. a** Metal-ferroelectric-metal configuration (lateral connection). **b** I–V characteristics of Ag/form I-crystal/Ag and Ag/form II-crystal/Ag.

**Differential scanning calorimetry**. Differential scanning calorimetry (DSC) was carried out on a TA DSC-Q2000 instrument. Crystals were taken on a Tzero aluminum pans without crippling the lid and heated from room temperature to the selected temperature at different rates of heating and cooling.

**Microscopy**. The shape transformation of GN during heating and cooling was observed by using optical microscope (Q32634 Q-imaging microscope, Linkam) equipped with a temperature-controlled heating/cooling stage (THMS600-PS).

**Single crystal X-ray diffraction**. Single crystal diffraction data of forms I and II were collected on a Bruker APEX DUO diffractometer with monochromated Mo$K_\alpha$ radiation ($\lambda$ = 0.71073 Å) and Photon II detector equipped with Cobra cooling device (Oxford Cryosystems). The Bruker Apex 3[58] software was used for data collection, integration, scaling, and absorption corrections. The data reduction was performed using SAINT[59] and XPREP[60]. The data was corrected for absorption effects by using SADABS[61]. The structure determination of both forms and refinement, using the OLEX2 interface[62], was performed by using the full matrix least-squares method, based on $F^2$ against all reflections as implemented in SHELXL-2014/7[63]. The HFIX command in SHELX-TL[63] was used for fixing the hydrogen atoms bonded to carbon atoms. Geometrical calculations were done using PLATON[64] and PARST[65]. The graphics for publication were generated using Mercury 4.3.1[66] and OLEX2[60]. More crystallographic details are provided in Supplementary Table 1.

**Nanoindentation**. The Agilent G200 nanoindenter with an XP head and Berkovitch diamond indenter was used on both forms of GN. The experiments were performed by using the continuous stiffness method to a selected depth with a strain rate of 0.05 s$^{-1}$, an amplitude of 2 nm, and a frequency of 45 Hz[67]. Calibration of the indenter was performed by using Corning 7980 silica reference sample (Nanomechanics S1495-25). The modulus was measured between 200 and 500 nm.

**Mechanical testing**. Three-point bending tests on form I and II crystals of GN were performed by using SEMTester DAQ-linear (model 8000-0014, MTI Instruments) and a, ensile tester at 280 and 310 K respectively. A 5 N load cell and a three point bending apparatus with a 1.5 mm span was used. A crosshead speed of 0.05 mm min$^{-1}$ was used.

**Measurement of the crystal actuation force**. The force generated by form I crystals of GN during the phase transition to form II was directly measured using AE801 force sensor. The cantilever-based force sensor was composed of a single crystal n-type silicon sandwiched between ion-implanted p-type resistors. Details on the design and operating principles of the sensor are provided in ref. [28]. A mechanical stress due to deflection of the beam causes piezoresistive effect and

results in a proportional change of the output voltage. Calibration of the force sensor was performed by 10 μm incremental deflections while maintaining linear correlation between the deflection and the output voltage. The conversion between output voltage and deflection was then corrected, and Eq. 3 was used to calculate the respective force generated by deflection

$$\delta = \frac{Fs^2}{3EI} \qquad (3)$$

where $\delta$ is the maximum cantilever tip deflection, $F$ is the applied force, $s$ is the length of the cantilever, and $E$ is the shaft's Young's modulus. For the force sensor used here, $s$ = 5 mm, $E$ = 160 GPa, and $I$ = 2.67 × 10$^{-16}$ m$^4$. For measurement, form I crystal was carefully sandwiched between a rigid wall and the tip of the force sensor below the phase transition temperature.

**Actuator data and materials property plots**. The data for other actuators were retrieved and plotted from the CES Selector 2019, Granta Design Limited, Cambridge, UK, 2019[68]. The data of other actuating materials in the database are divided into over 20 classes depending on the available attributes.

**Frequency-dependent capacitance-voltage measurements (C–V)**. The frequency-dependent capacitance–voltage (C–V) and the conductance–voltage (G–V) measurements were carried out using Agilent B 1505 A curve tracer and manual probes. Figure 4a shows the four terminal pair configuration used in our measurements. For the measurements, crystals of GN (form I and form II separately) were placed between two metal plates, where Cu and Ag act as the bottom and top plates of the capacitor, respectively as depicted in Fig. 4b. The measurements were performed at room temperature.

**Equilibrium crystal structure of forms I and II of GN**. The experimental crystal structures for forms I and II of GN were subjected to periodic dispersion-corrected density functional theory (DFT-D) geometry optimization using CASTEP 8.0[69] as implemented in BIOVIA Materials Studio 8.0[70] using the PBE generalized gradient approximation (GGA) exchange-correlation functional[71] and norm-conserving pseudopotentials[72]. The D2 (G06) semi-empirical dispersion correction of Grimme[73] was used. Brillouin zone integrations were performed on a symmetrized Monkhorst–Pack k-point grid with a separation of 0.07 Å$^{-1}$. The plane-wave basis set cut-off was set at 750 eV. The BFGS[74] algorithm was used for the geometry optimization. The structural optimization was considered complete when the following convergence criteria were satisfied: maximum energy change of 1 × 10$^{-5}$ eV per atom, maximum force of 3 × 10$^{-2}$ eV Å$^{-1}$, maximum stress of 5 × 10$^{-2}$ GPa, and maximum displacement of 1 × 10$^{-3}$ Å. The periodic DFT-D geometry optimization was performed first at fixed volume by optimizing only the atomic positions. This was followed by a second geometry optimization that relaxed both atomic positions and unit cell parameters. The resulting equilibrium structures for I and II were used as the basis for further calculations as detailed below.

**Energetic barrier for cell transformation from form I to form II**. The CASTEP generated equilibrium crystal structure for I was used to generate a series of strained crystals. The strained crystals were generated by incrementally changing the cell lengths and β angles starting with the equilibrium cell geometry of form I and gradually transforming the cell parameters to those found in the CASTEP-generated equilibrium crystal structure. A total of 6 crystals were generated: the strain-free form I equilibrium structure (strain step 0), 4 strained cells that represent the intermediate cell geometries during the martensitic phase transition and a sixth strained cell that has the equilibrium cell geometry found in the equilibrium form II structure. All 6 crystal structures were used as input for fixed-volume geometry optimization in VASP[75–77] in order to estimate the strain energy in the GN crystal as it rapidly switches between forms I and II. VASP uses the projector-augmented wave (PAW)[78] method with plane-wave basis sets and PAW pseudopotentials. In all calculations, the PBE[71] GGA was used for the exchange-correlational functional coupled with the D3 dispersion-correction[79]. By using the cif2cell[80] program, VASP input files were generated. A tight K-point mesh with maximum K-point distance set to 2π × 0.032 Å$^{-1}$ was generated by using a Γ-centered Monkhorst–Pack scheme. 700 eV was the cut-off energy for the plane-wave basis and the convergence threshold was set at 1 × 10$^{-4}$ eV. The geometry was considered converged when all forces were below 0.03 eV Å$^{-1}$ for each self-consistent filed cycle. The strain energy (ΔE$_s$) for each cell is reported relative to the equilibrium energy of the strain-free I crystal (strain step 0) as reported by VASP using the same convergence thresholds reported above.

**Simulation of cooperative rotational energy barrier in GN**. In order to estimate the energy barrier for cooperative motion of the GN ions during the martensitic phase transition, a bilayer of hexagonal hydrogen-bonded sheets of GN ions were extracted from the equilibrium form I structure. All the ions in the top layer were rotated incrementally relative to the bottom layer at increments of 10°. The rotation angle is denoted ϕ. At each step, the energy of the bilayer ring topology was estimated using VASP. The VASP control parameters and convergence thresholds were the same as those outlined above. The rotational energy barrier (ΔE$_{rot}$)

reported is relative to the energy of the ring topology found in the equilibrium form I structure ($\phi = 0°$).

**Reporting summary**. Further information on research design is available in the Nature Research Reporting Summary linked to this article.

## Data availability

The single crystal diffraction data generated in this study have been deposited in the Cambridge Crystallographic Data Center (CCDC) under accession codes 2123337 and 2123338. All data are available from the corresponding author upon request.

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

## Acknowledgements

We thank New York University Abu Dhabi for the financial support for this work. This research was partially carried out using the Core Technology Platform resources at New York University Abu Dhabi. The theoretical calculations were performed using the high-performance computing facilities at Khalifa University.

## Author contributions

P.N. and D.P.K. conceived the study. D.P.K. and R.F. prepared the crystals. D.P.K., E.A., and R.F. performed the mechanical tests. D.P.K. performed the thermal analysis, phase transition experiments, structural analysis, thermal expansion analysis, nanoindentation, and microscopy. G.D., L.C., and D.P.K. performed the electrical measurements. J.M.H. and D.P.K. performed the force measurements, and J.M.H. processed the data and prepared the plots. D.P.K. and P.N. analyzed the structural change and established the mechanism. D.P.K., Z.A., and I.T. recorded the videos and D.P.K. processed the data. S.M. performed the theoretical calculations. D.P.K. prepared the figures for publication. The paper was written with the contributions from D.P.K., G.D., J.H.M., S.M., and P.N. All authors have given approval to the final version of the paper.

## Competing interests

The authors declare no competing interests.
