## [Peer review file · Nature Communications]

REVIEWER COMMENTS

Reviewer #1 (Remarks to the Author):

In this manuscript, the authors report on an organic crystal of guanidinium nitrate (GN) with exceedingly large strokes (~51%). As the authors have noted, the crystal can reversibly and effectively convert energy into mechanical work around room temperature with work density. The presented work is important and interesting from a scientific perspective. The experimental methodology is appropriate and thorough, and includes detailed crystallographic structural analysis along with thermophysical and theoretical analysis. The proposed mechanisms for structural changes are reasonable. The manuscript is very well written and easy to follow. I recommend the acceptance of this manuscript after minor revision according to the following suggestions.

- 1) The authors discussed the current efforts in the field of dynamic crystals, namely the preparation of crystals which can transform rapidly and efficiently. However, the controllability over the crystal phase transition and the (possibly) corresponding shape deformation is also of crucial importance for practical applications, which should be discussed and highlighted.
- 2) To better characterize the phase transition and thermal expansion process of guanidinium nitrate crystals, a figure panel to highlight the temperature-dependent unit-cell parameters should be provided in the main manuscript.
- 3) The crystal system appears to have a significant thermal expansion. The authors might take advantage of the SCXRD data to calculate the thermal expansion properties and coefficients.
- 4) Guanidinium nitrate has an elasticity and hardness larger than those of most polymers, gels and elastomers. The maximum force density of guanidinium nitrate crystals indeed looks higher than many actuators. The authors are suggested to provide the data or movies to illustrate elasticity through the kinetic energy generated by the crystal deformation.

Reviewer #2 (Remarks to the Author):

In this manuscript, the authors present a case study of the single crystal of guanidinium nitrate that exhibits extremely high linear stroke, which could be potentially a very important discovery, which is why the results should be interpreted extremely carefully. If the observations are real and acceptable, this work should be regarded as an important supplement to the field of mechanical properties of

organic crystals. Besides, it will also arouse great interest of the science community, thus the topic is suitable for publication in Nature Communications.

However, there are some experimental details and scientific claims that might require further attention and thorough assessment, and I also feel that the overall quality of the manuscript could be technically improved further. I, therefore, cannot recommend it for publication at this stage yet a major revision is required. I hope to get the authors' kind understanding and would be happy to discuss again if the manuscript is given the opportunity to be revised. I listed major/minor issues below and I hope they can help authors to improve the manuscript.

Major issues:

(1) The authors have not stated whether this system is designed on purpose or discovered by accident. It seems like a very special system to me. Can the mechanism described in this paper guide the future design of other organic actuators? Please kindly explain.

(2) There is no problem that the guanidinium nitrate is an organic compound. However, it is the kind of organic compounds which are closer to the inorganic compounds. Therefore, can similar observations be found in a wider range of organic compounds? In addition, the crystals of guanidinium nitrate are ionic crystals composed of the molecular ions. It seems unreasonable that organic molecular crystals are mainly introduced in the introduction part.

(3) I didn't find the relevant expressions on how the values of linear stroke were calculated in the article or the supporting information. Also, please kindly describe the uncertainty of the values.

(4) Please kindly describe the crystal plane identification process of the two crystal forms and mark the crystal planes on the crystals in the photomicrographs.

(5) A very recent work on dynamic crystals should be noted (CrystEngComm, 2021, 23, 6838-6842, not cited), which provides a very large number of statistics on previous reports (Table S5 in supporting information) to highlight the advance. However, I found that Table S3 and Table S4 in this work are quite incomplete. There should be a lot of reports on thermal expansion on crystals (Prof. Len Barbour's series of work, for example), but only a few of them are listed in the table. If the authors want to state this work sets the highest stroke value with reversible characteristics of an organic single crystal actuator, all relevant work has to be counted and listed in the tables for a better comparison.

(6) Line 131: the [102] direction stated in this line seems incorrect. Please kindly check.

(7) It should be necessary to provide proper guidance for the follow-up design on this issue in the conclusion part.

Minor issues:

(1) The "exceptionally high work density" in the title is a very subjective description and should be revised.

- (2) Please kindly add scale bars to Figure S1 and Figure S8 in the supporting information.
- (3) A classic review on dynamic crystals by Prof. Sato should be cited (Nat. Chem. 2016, 8, 644-656).
- (4) Not all authors listed in the reference 1, 10, 12, 19, 26, 27, 30, 34, 47, 56. Kindly check.

Overall, I think this is a very nice piece of work and I think the authors have done some outstanding experimental work. However, only with the treatment of the above issues to the satisfaction of the Editor, I recommend publication in Nature Communications.

Reviewer #3 (Remarks to the Author):

In this manuscript, the authors report a Dynamic organic crystal (GN) that could exert a linear stroke of 51%, which is novel. Although the experimental results are good, some aspects need to be corrected before they can be accepted:

1, the theoretical understanding by DFT-D is very weak, they only mention that “DFT-D simulations reveal that the stacking of the ions in forms I and II are equienergetic”, however, the details of the calculations are not given anywhere in the manuscript or SI. We even don't know which exchange function they are using and whether vdw is considered. Besides, it would be better to do an NEB calculation to show the energy barrier for the forms I to II transition to show the energy needed to activate this transition.

2, the authors show that the crystal could change the habit plane during cooling (Figure 1) while a straight transition happens for the heating process, can the authors have a more in-depth discussion for this phenomenon? Maybe DFT calculations of the surface energies would give some insights.

3, In figure 3, various actuation materials are compared but somehow shape memory alloy is not considered, can the authors add a comparison to that as well.

4, from figure 4, these two forms show distinct ferroelectric properties, which is interesting. The molecular or structural aspects should be discussed. Would be interesting to see the P-E loop measurements and DFT calculations of the net polarizations for the two forms.

Overall, the experimental results are impressive while the theoretical understanding is, by far, very disappointing.

Response to the comments from the reviewers

We thank all reviewers for the valuable comments which have contributed to improve the quality of our manuscript. We considered all the comments, and we tried to address them to the best of our ability. Unless stated otherwise, all numbers of the figures and supplementary materials refer to the *revised* version of the manuscript. Together with this submission we have also provided marked copies of the main text and the supplementary materials where the changes to the original version have been marked. For convenience, in the PDF version of this response the original comments from the reviewers are highlighted in *blue color*, our response is provided in black color, and the text that was modified or added to the manuscript is marked with *red color*.

Response to the comments from Reviewer #1

Comment: *In this manuscript, the authors report on an organic crystal of guanidinium nitrate (GN) with exceedingly large strokes (~51%). As the authors have noted, the crystal can reversibly and effectively convert energy into mechanical work around room temperature with work density. The presented work is important and interesting from a scientific perspective. The experimental methodology is appropriate and thorough, and includes detailed crystallographic structural analysis along with thermophysical and theoretical analysis. The proposed mechanisms for structural changes are reasonable. The manuscript is very well written and easy to follow. I recommend the acceptance of this manuscript after minor revision according to the following suggestions.*

Response to the comment: We thank the Reviewer for their generally positive assessment of the work and the valuable suggestions. We appreciate that the Reviewer recognizes the relevance and the impact of the results presented in the manuscript. We share the Reviewer's sentiment and hope that the results presented here will open up new directions for applying simple organic molecular crystals as energy-harvesting materials at room temperature. All suggestions from the Reviewer have been considered and addressed in the revised version of the manuscript and the supplementary materials.

Comment: *1) The authors discussed the current efforts in the field of dynamic crystals, namely the preparation of crystals which can transform rapidly and efficiently. However, the controllability over the crystal phase transition and the (possibly) corresponding shape deformation is also of crucial importance for practical applications, which should be discussed and highlighted.*

Response to the comment: We thank the Reviewer for this comment and we agree that the controllability of the crystal phase transition and the corresponding shape deformation is one of the most important aspects for practical applications of adaptive crystals. Indeed, realizing practical applications by using dynamic crystals poses challenges with collective molecular movement and the related lack of control over the mechanical response; the molecular movements are hardly controlled once they are initiated. Most of the reported compounds show

actuation above room temperature and in many cases the thermal hysteresis between the forward and reverse transition is significant. On the other hand, with respect to the shape transformation, some of the reported dynamic crystals show disintegration, splintering and deformation during the phase transition from one form to another, which significantly hampers the implementation of these materials in devices for reversible operation. In this work, we report that GN undergoes a phase transition around room temperature while it also generates unexpectedly large linear stroke without deterioration of crystal integrity. This combination of properties is unique, and turns this material into a promising and potential candidate for real-world practical applications.

To respond to the Reviewer's comment, the following text has been added to the manuscript with the relevant references:

“Since the molecular movements are hardly controlled once they are initiated, realization of practical applications by using dynamic crystals poses challenges with the collective molecular movement and the related lack of control over the mechanical response. With regards to the shape transformation, some of the reported dynamic crystals are prone to disintegration, deterioration, splintering and deformation during the phase transition, which hampers significantly the implementation of these materials into devices with cyclic operation.^{31–37}”

31. Tamboli, M. I.; Karothu, D. P.; Shashidhar, M. S.; Gonnade, R. G. & Naumov, P. Effect of crystal packing on the thermosalient effect of the pincer-type diester naphthalene-2,3-diyl-bis(4-fluorobenzoate): A new class II thermosalient solid. *Chem. Eur. J.* 24, 4133–4139 (2018).

32. Colin-Molina, A. *et al.* Thermosalient amphidynamic molecular machines: Motion at the molecular and macroscopic scales. *Matter* 1, 1033–1046 (2019).

33. Panda, M. K. *et al.* Strong and Anomalous Thermal Expansion Precedes the Thermosalient Effect in Dynamic Molecular Crystals. *Sci. Rep.* 6, 29610 (2016).

34. Takeda, T., Ozawa, M. & Akutagawa, T. Jumping crystal of a hydrogen-bonded organic framework induced by the collective molecular motion of a twisted π system. *Angew. Chem. Int. Ed.* 58, 10345–10352 (2019).

35. Smets, M. M. H. *et al.* Understanding the solid-state phase transitions of dl-norleucine: An In Situ DSC, Microscopy, and Solid-State NMR Study. *Cryst. Growth Des.* 15, 5157–5167 (2015).

36. Seki, T.; Mashimo, T.; Ito, H. Anisotropic strain release in a thermosalient crystal: correlation between the microscopic orientation of molecular rearrangements and the macroscopic mechanical motion. *Chem. Sci.* 10, 4185–4191 (2019).

37. Sahoo, S. C. *et al.* Kinematic and mechanical profile of the self-actuation of thermosalient crystal twins of 1,2,4,5-tetrabromobenzene: A molecular crystalline analogue of a bimetallic strip. *J. Am. Chem. Soc.* 135, 13843–13850 (2013).

Comment: 2) To better characterize the phase transition and thermal expansion process of guanidinium nitrate crystals, a figure panel to highlight the temperature-dependent unit-cell parameters should be provided in the main manuscript.

Response to the comment: We thank the Reviewer for this comment. We have already included a plot that shows the changes of the unit cell parameters during heating (starting from 100 K) as well as around the phase transition. We have now included a more detailed figure

(Figure 2d) showing the reversible changes during the thermal phase transition. The figure was enhanced to highlight the changes in the unit cell in panel a.

Figure 2 | Changes in the crystal structure and crystal size and shape during the phase transition. (a) Comparison of the unit cells before and after the phase transition. The values of the characteristic unit cell parameters in the two phases are given. The doubling of the c axis that occurs during the transition from form I to II is now shown here. (b,c) Cartooned crystal habits of form I (b; 250 K) and form II (c; 310 K) shown with molecular orientation in form I (b) and form II (c) crystals viewed in the [001] and [100] directions. Upon heating from 290 K to 310 K, the

distance between the adjacent sheets containing nitrate and guanidinium ions increases from 3.605 Å to 3.749 Å while the adjacent hexameric layers are apparently twisted in respect to each other. (d) Changes in the unit cell parameters of form I before and during the reversible phase transition. (e) Cyclability of a single crystal over the phase transition monitored by the change in its length. The expansion and shrinkage of the crystal were recorded over 20 thermal cycles.

Comment: 3) *The crystal system appears to have a significant thermal expansion. The authors might take advantage of the SCXRD data to calculate the thermal expansion properties and coefficients.*

Response to the comment: We agree with the Reviewer that this material shows a significant thermal expansion. We have already included the thermal expansion coefficients in the main manuscript. For convenience, the section where we discuss the thermal expansions is copied below.

“To investigate the change of the structure before the phase transition, the temperature-dependent variation in the unit cell parameters was recorded (Figure 2C; Supplementary Table 2). The expansivities along the principal axes of form I prior to the phase transition and the respective expansivity indicatrix are available from Supplementary Figure 5. The thermal expansion of the crystal of form I preceding the transition to form II is strongly anisotropic. The minor (X1), medium (X2) and major (X3) principal axes in form I are oriented along the [102], [010] and [001] directions, respectively. The crystal undergoes small biaxial negative expansion, with coefficients of $-4.98 \times 10^{-6} \text{ K}^{-1}$ and $-6.81 \times 10^{-6} \text{ K}^{-1}$, and a large (‘colossal’) uniaxial positive thermal expansion (PTE) parallel to *c* with coefficient of $211.46 \times 10^{-6} \text{ K}^{-1}$. The volumetric thermal expansion (VTE) is $210.79 \times 10^{-6} \text{ K}^{-1}$. The positive axial and volumetric thermal expansion coefficients of form I are comparable to or exceed those reported for other compounds (Supplementary Table 3).^{12,38–42}”

Comment: 4) *Guanidinium nitrate has an elasticity and hardness larger than those of most polymers, gels and elastomers. The maximum force density of guanidinium nitrate crystals indeed looks higher than many actuators. The authors are suggested to provide the data or movies to illustrate elasticity through the kinetic energy generated by the crystal deformation.*

Response to the comment: Our understanding is that in this comment the Reviewer is referring to the Young’s modulus through a force measurement as an alternative method to nanoindentation. In order to determine the stiffness of the crystal the following equation was used

$$F = k\delta$$

where *F* is the absolute maximum force generated by the crystal or blocking force at zero displacement, *k* is the stiffness of the crystal, and δ is the maximum unrestrained displacement generated with the absence of a load.

The force being measured by the force sensor here is the effective force expressed as

$$F_{\text{eff}} \approx F_{\text{max}} \left(\frac{k_L}{k_C + k_L} \right)$$

where $k_L = 5.13 \text{ N/m}$ is the stiffness of the load (in this case, the sensor). We have not measured F_{max} which would be the blocking force output when the crystal is not able to generate any displacement. That would require a load with varying stiffness that can be modulated to reach a stiffness high enough such that the crystal cannot displace it anymore. Only then one can calculate the real stiffness of the crystal from which the effective Young's modulus of the entire structure can then be calculated.

We also would like to note that we do not recommend the use of this method, given that the contact area between the crystal and the sensor is not always perfectly rectangular, and can change throughout the actuation as small parts of the crystal get compressed under high loads. Also, the contact angle between the sensor and the crystal can change to become larger or lower than 90 degrees which then introduces small bending effects, in addition to the uniaxial compression in the crystal, and can further complicate the calculations.

Response to the comments from Reviewer #2

Comment: *In this manuscript, the authors present a case study of the single crystal of guanidinium nitrate that exhibits extremely high linear stroke, which could be potentially a very important discovery, which is why the results should be interpreted extremely carefully. If the observations are real and acceptable, this work should be regarded as an important supplement to the field of mechanical properties of organic crystals. Besides, it will also arouse great interest of the science community, thus the topic is suitable for publication in Nature Communications. However, there are some experimental details and scientific claims that might require further attention and thorough assessment, and I also feel that the overall quality of the manuscript could be technically improved further. I, therefore, cannot recommend it for publication at this stage yet a major revision is required. I hope to get the authors' kind understanding and would be happy to discuss again if the manuscript is given the opportunity to be revised. I listed major/minor issues below and I hope they can help authors to improve the manuscript.*

Response to the comment: We thank the Reviewer for their comments, and we are delighted to hear words of appreciation of our contributions to this rapidly growing research field. We also thank the Reviewer for recognizing the originality of the results presented in this manuscript. We find their comments on the manuscript to be very helpful in further improving the quality of its contents. Correspondingly, all suggestions have been considered and addressed in the revised version, and we hope that this is to the satisfaction of the Reviewer. We would appreciate to have further comments in case we missed anything.

Comment: *Major issues:*

(1) The authors have not stated whether this system is designed on purpose or discovered by accident. It seems like a very special system to me. Can the mechanism described in this paper guide the future design of other organic actuators? Please kindly explain.

Response to the comment: We confirm that we have discovered this material accidentally, during our more general search for organic crystalline materials with thermosalient, actuating, and other dynamic properties. As part of that broader research, we have synthesized and analyzed a number of derivatives of guanidine. While most of these derivatives did not show thermosalience or other dynamic behavior, a few did exhibit interesting properties, and the material reported here was particularly interesting to us for its extraordinary expansion upon phase transition. We believe that designing materials with desired dynamic properties is a far-reaching goal, and requires a lot of effort to be realized in the future.

To respond to the Reviewer's comment, we have included the following sentences in the revised manuscript:

“During our broader screening for materials with thermosalient, actuating, and other dynamic properties we have synthesized and analyzed a number of guanidine derivatives. While most of these derivatives did not show thermosalience or other dynamic behavior, a few were found to exhibit interesting properties. The material reported here was particularly interesting for its extraordinary expansion upon heating.”

Comment: *(2) There is no problem that the guanidinium nitrate is an organic compound. However, it is the kind of organic compounds which are closer to the inorganic compounds. Therefore, can similar observations be found in a wider range of organic compounds? In addition, the crystals of guanidinium nitrate are ionic crystals composed of the molecular ions. It seems unreasonable that organic molecular crystals are mainly introduced in the introduction part.*

Response to the comment: We thank the Reviewer for this comment. To the best of our knowledge, and based on our experience in the past, large linear actuation such as the one we demonstrate in this work has not been observed for an organic or inorganic compound yet. Only six ionic compounds (inorganic ionic and organic ionic) show dynamic behavior during a phase transition: oxitropium bromide,¹ helquat salt,² phenylalkyl halides,³ tetrabutyl-*n*-phosphonium tetraphenylborate,⁴ a nickel complex with ethylenediamine and oxalate,⁵ and an organic-inorganic hybrid copper complex ((Himd)₂[CuCl₄] (Himd⁺ = imidazolium)).⁶ However some of these compounds disintegrate during the reversible phase transition and also the actuation is less pronounced. To the best of our knowledge, guanidinium nitrate shows by far the most pronounced actuation between the two phases during a phase transition from the known organic, inorganic or hybrid crystalline materials.

1. Skoko, Ž., Zamir, S., Naumov, P. & Bernstein, J. The thermosalient phenomenon ‘jumping crystals’ and crystal chemistry of the anticholinergic agent oxitropium bromide. *J. Am. Chem. Soc.* 132, 14191–14202 (2010).

2. Nath, N. K. *et al.* Single-crystal-to-single-crystal transition in an enantiopure [7] helquat salt: the first observation of a reversible phase transition in a helicene-like compound. *Chem. Eur. J.* 21, 13508–13512 (2015).
3. Nitta, I., Watanabe, T., & Taguchi, I. *Bull. Chem. Soc. Jpn.* 34, 1405 (1961).
4. Takamizawa, S. & Takasaki, Y. Shape-memory effect in an organosuperelastic crystal. *Chem. Sci.* 7, 1527–1534 (2016).
5. Sato, O. Dynamic molecular crystals with switchable physical properties. *Nat. Chem.* 8, 644–656 (2016).
6. Wang, X.-L. *et al.* Giant Single-Crystal Shape Transformation with Wide Thermal Hysteresis Actuated by Synergistic Motions of Molecular Cations and Anions. *Chem. - Eur. J.* 26, 6778–6783 (2020).

The following sentences have been added to the revised manuscript with relevant references to clarify this point:

“Indeed, several recent examples of mechanical effects based on phase transitions in organic crystals, inorganic crystals, hybrid crystals and a few ionic compounds (inorganic ionic and organic ionic) have indicated the hidden potential of these materials for fast actuation.”^{22–45}

22. Skoko, Ž., Zamir, S., Naumov, P. & Bernstein, J. The thermosalient phenomenon ‘jumping crystals’ and crystal chemistry of the anticholinergic agent oxitropium bromide. *J. Am. Chem. Soc.* 132, 14191–14202 (2010).
23. Nath, N. K. *et al.* Single-crystal-to-single-crystal transition in an enantiopure [7] helquat salt: the first observation of a reversible phase transition in a helicene-like compound. *Chem. Eur. J.* 21, 13508–13512 (2015).
24. Nitta, I., Watanabe, T., & Taguchi, I. *Bull. Chem. Soc. Jpn.* 34, 1405 (1961).
25. Takamizawa, S. & Takasaki, Y. Shape-memory effect in an organosuperelastic crystal. *Chem. Sci.* 7, 1527–1534 (2016).
26. Sato, O. Dynamic molecular crystals with switchable physical properties. *Nat. Chem.* 8, 644–656 (2016).
27. Wang, X.-L. *et al.* Giant Single-Crystal Shape Transformation with Wide Thermal Hysteresis Actuated by Synergistic Motions of Molecular Cations and Anions. *Chem.–Eur. J.* 26, 6778–6783 (2020).

Comment: (3) *I didn't find the relevant expressions on how the values of linear stroke were calculated in the article or the supporting information. Also, please kindly describe the uncertainty of the values.*

Response to the comment: We note that we have already provided the procedure for calculating the linear strokes in the Methods section. The strokes under loading were measured directly, by using a piezoelectric force sensor with an instrument error around 0.5%. The sensor is essentially a n-type silicon-based cantilever beam sandwiched between ion-implanted P-type resistors. Any deflection in the beam driven by the crystal expansion causes mechanical stress in the beam which is in turn registered as a proportional voltage output due to the piezoelectric effect in the resistors. The sensor was calibrated before every experiment by applying a controlled displacement using an automated micrometer stage and the linearity between output voltage and imposed displacement was ensured.

For convenience, we have copied the section describing these details below:

Measurement of the crystal actuation force. The force generated by form I crystals of guanidine nitrate during phase transition to form II was directly measured using AE801 force sensor. The force sensor is comprised of a single crystal n-type silicon and has one ion-implanted p-type resistor on each side (complete instrumental details for the force sensor measurements are available in ref.²⁹). The deflection of the beam induces mechanical stress that causes a change in the output voltage through the piezoresistive structure. The sensor was first calibrated by performing deflections at 10 μm increments while ensuring linearity between output voltage and the deflection. The conversion between output voltage and the deflections was then corrected and Eq. 3 was used to calculate the respective force generated by deflection, δ

$$\delta = \frac{Fs^2}{3EI} \quad (\text{Eq. 3})$$

where δ is the maximum tip deflection of the cantilever, F is the applied force causing the deflection, s is the length of the cantilever (assuming the contact point with the sensor is at the end of the cantilever) and E is the shaft's Young's modulus. For the specific sensor used in this work, $s = 5 \text{ mm}$, $E = 160 \text{ GPa}$, and $I = 2.67 \times 10^{-16} \text{ m}^4$. The crystal was sandwiched between a rigid wall and the tip of the force sensor and lightly pressed to ensure that none of the output force is lost in the recording. This is also to make sure that the point of contact with the sensor is always perpendicular to the cantilever since a change in the angle can affect the force measurements.”

We would also like to note that the average stroke reported from 6 GN crystals is 0.863 ± 0.103 mm. However, we provided stroke as a range from 0.35-0.95 mm since the stroke largely depends on the dimensions of the crystal. To clarify this point, the following sentence was added to the revised manuscript:

“The strokes measured from 6 crystals were in the range 0.35-0.95 mm, since the stroke largely depends on the actual dimensions of the crystal.”

Comment: (4) Please kindly describe the crystal plane identification process of the two crystal forms and mark the crystal planes on the crystals in the photomicrographs.

Response to the comment: We have included the procedure used for face indexing in the Methods section, and we have provided illustrations of the crystal habits with indexed faces in the Supporting Information.

The following text was added in the revised manuscript:

“A series of images were then taken with a video microscope as the crystal (form I and form II) was rotated 360° about the ϕ -axis. Stepping through the images, the T-tool in the Bruker APEX3 plug-in for face indexing was used to define all visible faces within the maximum Miller index range.”

The following image has been added to the Supporting Information:

Supplementary Figure 5. Face indexing of form I (A) at 200 K and form II (B) at 305 K.

Comment: (5) A very recent work on dynamic crystals should be noted (*CrystEngComm*, 2021, 23, 6838-6842, not cited), which provides a very large number of statistics on previous reports (Table S5 in supporting information) to highlight the advance. However, I found that Table S3 and Table S4 in this work are quite incomplete. There should be a lot of reports on thermal expansion on crystals (Prof. Len Barbour’s series of work, for example), but only a few of them are listed in the table. If the authors want to state this work sets the highest stroke value with reversible characteristics of an organic single crystal actuator, all relevant work has to be counted and listed in the tables for a better comparison.

Response to the comment: We thank the Reviewer for bringing to our attention the desolvation-induced crystal jumping of spironolactone–saccharin cocrystal work, and the extensive list of thermal expansion data provided in that work. We would like to clarify here that the thermal expansion when the material does not undergo a phase transition and the expansion or contraction (of the crystal length) during a phase transition (upon heating and cooling) are two very different phenomena, and they cannot be compared directly. Several examples of dynamic effects based on phase transitions have shown that the typical strokes (actuation) that can be obtained rarely exceed several percent, and are typically less than 12%. The single crystals of guanidinium nitrate undergo rapid and reversible first-order phase transition around room temperature and exhibit a stroke of 51%, which is the highest value reported for transitioning between two phases. Perhaps coincidentally, the thermal expansion of the crystal of form I preceding the transition to form II is strongly anisotropic. The crystal undergoes small biaxial negative expansion with coefficients of $-4.98 \times 10^{-6} \text{ K}^{-1}$ and $-6.81 \times 10^{-6} \text{ K}^{-1}$, and a large ('colossal') uniaxial positive thermal expansion (PTE) parallel to *c* with a coefficient of $211.46 \times 10^{-6} \text{ K}^{-1}$. The volume thermal expansion (VTE) of the material is $210.79 \times 10^{-6} \text{ K}^{-1}$. We note that the positive axial and volumetric thermal expansion coefficients of form I are comparable to or exceed those reported for other compounds (Supplementary Table 3).

To respond to the Reviewer's comment, we have now included revised Supplementary Tables S3 and S4 with updated list of compounds.

Table S3. Selected coefficients of uniaxial thermal expansion (TE) and their volumetric thermal expansion (VTE) reported.

Compound	Temp range (K)	Uniaxial TE $\alpha_x (10^{-6} \text{ K}^{-1})$	VTE $\alpha_v (10^{-6} \text{ K}^{-1})$	Reference
(phenylazophenyl)Pd(hexafluoroacetate)	220-350 100-270	260.4, 39.4, -79.9 124.0, 105.4, 114.9	247.8 (α form) 255.5 (β form) 213.13	14
18-crown-6-nitromethane	180-273	X1 -129(15) X2 144(14) X3 282(16)	311	15
4-aminobenzonitrile and 4-(dimethylamino) benzonitrile in 1:2	100-300	X1 24.3 X2 90.9 X3 105	222	16
1,2,3,4-cyclobutane-tetracarboxylic acid and 4,4'-bipyridylethylene	120-298	X1 4(5) X2 25(4) X3 147(8)	183	17
4-phenylazopyridine 4,6-dCl and 4,6-dichlororesorcinol in 2:1 ratio	290-260	X1 -116 X2 29 X3 316	229	18
FMOF1	90-295	230	300	19
4-aminobenzonitrile (ABN)	180-300	X1-24.8 X2- 61.3, X3-138.2	225.1	20

triphenylethenyl isocyanide complex	gold	123-298	X1 -285.2, X2 22.1, X3 579.1,	315.3	21
1D CP of Pb(II)		298-398	X1 -53.7, X2 63.2 X3 189.9	200.7	22
Pb(II) [Pb(SCN) ₂ (2F-spy) ₂]		173- 393	X1 6.3, X2 35.2, X3 93.4	136.4	23
[Zn(benzoate)(25F-spy) ₂]		303-413	X1 -54.46, X2 106.42, X3 151.69	204.26	24
[Cu ₂ (benzoate) ₄ (L) ₂], where L = 4-styrylpyridine (4spy)		273-473	X1 13.9, X2 56.0, X3 166.3	241.8	25
2'-fluoro-4-styryl-pyridine (2F- 4spy) (2)		273-473	X1 21.8, X2 38.3, X3 167.7	233.1	25
3'-fluoro-4-styryl-pyridine		273-473	X1 -13.8, X2 64.5, X3 228.3	285.6	25
MCF-82 1		112-300	X1 61, X2 482, X3 -218	319	26
(S,S)-octa-3,5-diyn-2,7-diol		225-330	156< α a>516, -32< α b>-85, -48< α a>-204		27
Olefin-I and several derivatives		190-250	X1 -2, X2 68, X3 128	194	28
Form I IMACET		298-373	α_a 225.9 α_b 238.8 α_c -290.0	181.2	29
4PAzP		260-290	X1 -116, X2 29, X3 316	328	30
4,4'-AP and 4,6-diX-res and other derivatives		190-290	X1 2, X2 17, X3 164	185	31
{[FeTp(CN) ₃] ₂ Co(Bib) ₂ }.5 H ₂ O		180-240	X1 85, X2 278, X3 1089	1498	32
ABN-2DMABN		100-300	α_{x1} 24.3, α_{x2} 90.9, α_{x3} 105.0	222	33
DAN-CA		85-385	27< $\alpha\alpha$ < 33, 56 < $\alpha\beta$ < 78,		34

		$35 < \alpha c < 58$		
24DNAN, the β -form	100-261	X1 -31(6), X2 -15(7), X3 262(20)	216	35
$\{[Zn(BTC)(HBPP)] \cdot H_2O\}_n$ (1·H ₂ O)	260-100	a, b and c are -6.0(1), 53.0(7) and -7.9(5)	39	36
(Himd) ₂ [CuCl ₄]		$\alpha_{a'} = -38 \times 10^{-6} \text{ K}^{-1}$, $\alpha_{b'} = 568 \times 10^{-6} \text{ K}^{-1}$, and $\alpha_{c'} = -184 \times 10^{-6} \text{ K}^{-1}$	346	37
IMD-HBC	100-360	X1 -115, X2 18, X3 210	110	38
Trianglimine form II	Cooling	X1 -17(1), X2 = 28(1), X3 = 145(5)	161	39
Trianglimine form III	Cooling	X1 -5(2), X2 59(1), X3 = 98(2)	151	39
GN	100-294	X1 -4.98, X2 -6.81, X3 211.46	210.79	Current work

Table S4. Reported crystals and their axial thermal expansions in percentage

Crystal	Maximum Axial Thermal Expansion	References
[Ni ^{II} (ethylenediamine) ₃](oxalate anion) complex	5%	Nat. Chem. 2014 , 6, 1079-1083.
Organic-inorganic hybrid Cu(II) complex, bis(imidazolium) tetrachlorocuprate, (Himd) ₂ [CuCl ₄]	~10%	Nat. Commun. 2019 , 10, 4805.
Cobalt (II) complex with a n -butyl group in its ligand, [Co(NO ₃) ₂ (L)]	6 – 7%	Nat. Commun. 2015 , 6, 8810.
Ag ₃ [Co(CN) ₆]	~6%	Science 2008 , 319, 794-797.
Pyridine-ICl and pyridine-IBr complexes	~2.7% and ~2.9%	CrystEngComm 2014 , 16, 237-243.
IMACET I-II (not reversible)	12.3%	Scientific Reports 2016 , 6, 29610.
Pentacene and Rubrene	~2.6% and ~2.3%	J. Phys. Chem. Letters 2012 , 3, 3325-3329.
α -(phenylazophenyl)palladium hexafluoroacetylacetonate	~4.6%	Nat. Commun. 2014 , 5, 4811.
7,7,8,8-tetracyanoquinodimethane- p -bis(8-hydroxyquinolinato)copper(II) (not reversible)	100%	J. Am. Chem. Soc. 2014 , 136, 590-593.

2,7-di([1,1'-biphenyl]-4-yl)-fluorenone	~10%	Nat. Commun. 2019 , 10, 4573.
Guanidinium nitrate (reversible)	51.8%	This work
naphthalene diimide (NDI) systems (crystal shrinks upon cooling)	-10 % (shrink)	J. Am. Chem. Soc. 2021 , 143, 5951–5957
[NiII(en)3](ox) complex (1)	-5 % (shrink)	Nat. Chem. 2014 , 6, 1079–1083.
TIPS-pentacene	10 %	Nat. Commun. 2018 , 9, 278.
N-[[4-p-dimethylaminophenylazo]benzoyl]-1-phenylethylamine [trans-(S)-1]	4.5% (along width)	Nat. Commun. 2018 , 9, 538.
[CoII(en)3](ox) (1, en=ethylenediamine; ox2) (crystal shrinks upon cooling)	-4.5% (shrink)	Angew. Chem. Int. Ed. 2017 , 56, 717–721.
1-d ₄ [(D ₄ BPTC)(azpy) ₂] _n (crystal shrinks upon cooling)	-4% (shrink)	Angew. Chem. 2016 , 128, 14848–14852

Comment: (6) Line 131: the [102] direction stated in this line seems incorrect. Please kindly check.

Response to the comment: We thank the Reviewer for this suggestion. We have rechecked the directions by using the program Pascal, and we can confirm that the reported directions are correct.

Comment: (7) It should be necessary to provide proper guidance for the follow-up design on this issue in the conclusion part.

Response to the comment: We feel that this research field is still at its infancy, and it is rather difficult to provide design strict principles for similar materials at this stage. However, some hints on the potential future directions that can be considered in order to search for similar materials can be provided as guidance. In order to provide some hints on this, we have included the following text in the conclusions section of the revised manuscript:

“In an attempt to verify the generality of the results, we have also synthesized and analyzed a number of other derivatives of guanidine. While most of these derivatives did not show thermosaliency or other dynamic behavior, a few of them did exhibit interesting dynamic properties. The material reported here was particularly interesting for its extraordinary expansion. We believe that designing materials with desired dynamic properties is a far-reaching goal and is likely to require a lot of systematic effort to be realized in the future.”

Comment: *Minor issues:*

(1) The "exceptionally high work density" in the title is a very subjective description and should be revised.

Response to the comment: We have considered this comment, and we have thoroughly discussed the suggested change. As authors of this work, we feel that the title has been selected to reflect a physical property which has been quantified in this work. The title, which might sound a little subjective, indeed reflects a value of a physically measurable quantity that has been compared to (physically measurable quantities) in other materials. That comparison has indeed shown that the measured values is exceptionally high, and therefore we would like to retain the title in its present form.

Comment: *(2) Please kindly add scale bars to Figure S1 and Figure S8 in the supporting information.*

Response to the comment: The scale bars have been added.

Comment: *(3) A classic review on dynamic crystals by Prof. Sato should be cited (Nat. Chem. 2016, 8, 644-656).*

Response to the comment: We have included this review in the reference section.

26. Sato, O. Dynamic molecular crystals with switchable physical properties. Nat. Chem. 8, 644–656 (2016).

Comment: *(4) Not all authors listed in the reference 1, 10, 12, 19, 26, 27, 30, 34, 47, 56. Kindly check.*

Response to the comment: We have rechecked all the references and reformatted them by using the NPG format.

Comment: *Overall, I think this is a very nice piece of work and I think the authors have done some outstanding experimental work. However, only with the treatment of the above issues to the satisfaction of the Editor, I recommend publication in Nature Communications.*

Response to the comment: We thank again the Reviewer for their thorough reading and the comments, and we hope that we have succeeded in satisfactorily addressing all of their suggestions.

Response to the comments from Reviewer #3

Comment: *In this manuscript, the authors report a Dynamic organic crystal (GN) that could exert a linear stroke of 51%, which is novel. Although the experimental results are good, some aspects need to be corrected before they can be accepted:*

Response to the comment: We thank the Reviewer for their generally positive assessment of the work and the very valuable suggestions. We also appreciate that the Reviewer recognizes the relevance and the impact of the results presented in this manuscript. All suggestions have been considered and addressed in the revised version of the manuscript and the supplementary materials.

Comment: *1, the theoretical understanding by DFT-D is very weak, they only mention that “DFT-D simulations reveal that the stacking of the ions in forms I and II are equienergetic”, however, the details of the calculations are not given anywhere in the manuscript or SI. We even don't know which exchange function they are using and whether vdW is considered.*

Response to the comment: We thank the Reviewer for their time and valuable feedback. Since the theoretical calculations were performed as support for understanding the mechanism for the phase transition, both the methodology and results for the theoretical work were placed in the Supporting Information document (SI) during our initial submission. However, in response to the Reviewer's feedback, we have moved the methodology for the theoretical calculations from the SI (previously pages 2-3 of SI) to the main text. We agree with the Reviewer that it would be better to have all the methodology in the main text.

Comment: *Besides, it would be better to do an NEB calculation to show the energy barrier for the forms I to II transition to show the energy needed to activate this transition.*

Response to the comment: We thank the Reviewer for their feedback. Following extensive molecular dynamics and DFT-D simulations, we found that the DFT-D model used in this work gives a reasonable compromise between accuracy and cost of the computational calculations. Moreover, the results from the calculations were consistent with the experimental data and provided important insight into the mechanism of the phase transition in GN. This insight was the fact that the energy required to deform the cell during the phase transition is compensated for by the energy released upon cooperative re-arrangement of the hydrogen bonded GN ions. This soft energy barrier for polymorphic phase transition facilitates the significant expansion of the crystal and hence the observed exceptional stroke of this system.

Comment: *2, the authors show that the crystal could change the habit plane during cooling (Figure 1) while a straight transition happens for the heating process, can the authors have a more in-depth discussion for this phenomenon? Maybe DFT calculations of the surface energies would give some insights.*

Response to the comment: We thank the reviewer for this comment. The transformation time (from one shape to another shape) for forward and backward reactions in GN are dependent on the rate of heating and cooling during phase transition. The forward reaction (heating) from form I to II is very fast and the transition completes in a few micro to milliseconds. However, from form II to form I (cooling), the transformation is slower than form I to II (heating). In figure 1e, form I to II (heating) was collected at a rate of 20 K/min while form II to form I (cooling) collected at a rate of 10K/min (figure 1f). The strain developed during the phase transition in a shorter time while heating at higher rates is responsible for the instantaneous transformation of the shape there by visually prohibiting the trapping of the bent shape near the phase boundary. Similar behavior is also observed during form II to form I transition (cooling) at a rate of 30 K/min. On the other hand the phase boundary with bent shape is observed during form I to form II transition at a heating rate of 2 K/min. All the transitions (with bent shape and without bent shape) have shown same mechanism during the heating and cooling. We have now included a supporting figure and a movie showing the effect of heating and cooling rate during phase transition. In the context of the theoretical calculations, modelling the habit changes observed in the crystal during the phase transition is beyond the tools available to us at present.

The following text was added to the revised manuscript:

“Both transitions depend on the rate of heating, and the phase boundaries were observed at different heating rates during the phase transitions (Supplementary Figure 4 and Supplementary Movie 3).”

Supplementary Figure 4. Effect of heating and cooling rate during phase transition. Form II is converted to form I during cooling (A–C) and form I is converted to form II during heating (D–E). The length of the scale bars in panels A–D is 400 μm and in panel D it is 600 μm .

Comment: 3, In figure 3, various actuation materials are compared but somehow shape memory alloy is not considered, can the authors add a comparison to that as well.

Response to the comment: We thank the Reviewer for this comment. We have already presented a comparison of guanidinium nitrate with a shape memory material (NiTi alloy). However, the comparison with other shape memory alloys could not be done due to the non-availability of the relevant data in the software that we use as a database of materials properties (Granta).

Comment: *4, from figure 4, these two forms show distinct ferroelectric properties, which is interesting. The molecular or structural aspects should be discussed. Would be interesting to see the P-E loop measurements and DFT calculations of the net polarizations for the two forms.*

Response to the comment: The accurate and more quantitative *P-E* loop may not be properly extracted directly from C-V measurements and a specific tool (Sawyer–Tower circuit) is required which is not available to the authors of this work. However, we thank the Reviewer for bringing this to our attention, and we will definitely consider such experiment in the continuation of this work with other promising derivatives. In the context of the theoretical calculations, we would like to note that modelling of the net polarizations of the two forms is beyond the scope of this work.

Comment: *Overall, the experimental results are impressive while the theoretical understanding is, by far, very disappointing.*

Response to the comment: We thank the Reviewer for their time and comments. In response to the Reviewer's comments, we have moved the theoretical methodology from the SI to the main text. The theoretical calculations are used only as a support to understand the rapid phase transition in GN crystals and as such we make only minimal references to this data in the main text. The relevant theoretical data is given in Figure 11 of the Supporting Information and is referred to where relevant in the main text.

REVIEWERS' COMMENTS

Reviewer #1 (Remarks to the Author):

In the revised version of the manuscript by addressing the reviewers' comments, the authors have provided stronger claims in support of the uniqueness of their material, and I believe that this manuscript could be acceptable for publication after another round of revision.

Concerning the controllability over the phase transition of molecular crystals, the authors are right in that “the phase transitions of molecular crystals are often hard to control once initiated, and in many cases the thermal hysteresis between the forward and reverse transition is significant. On the other hand, with respect to the shape transformation, some of the reported dynamic crystals show disintegration, splintering and deformation during the phase transition”. However, it should be acknowledged that the recent efforts in this direction have made some progress. By manipulating the phase boundaries, the thermal hysteresis of phase transition of the organic crystal can be greatly reduced to 2–3 °C, repeatable and reversible while retaining their crystalline nature (Small 2021, 17, 2006757). Moreover, a second-order phase transition can effectively decrease the rate of change in molecular configuration (J. Am. Chem. Soc. 1999, 121, 2808-2819), which might be also a direction towards the controllability over dynamic crystals. It would be greatly appreciated if these aspects can be discussed.

Reviewer #2 (Remarks to the Author):

I am very satisfied that the authors have now added enough reported data to the supporting information for comparison with this work. The authors have addressed all comments, and the manuscript could be accepted in its present form.

Reviewer #3 (Remarks to the Author):

The authors have almost ignored all my comments and suggestions, and the current theoretical understanding is still very weak. I would suggest rejecting this paper.

Response to the comments from the reviewers

We thank all reviewers for the valuable comments which have contributed to improve the quality of our manuscript. We considered all the comments, and we tried to address them to the best of our ability. Unless stated otherwise, any numbers of the figures and supplementary materials refer to the *revised* version of the manuscript. Together with this submission we have also provided marked copies of the main text and the supplementary materials where the changes to the original version have been marked. For convenience, in the PDF version of this response the original comments from the reviewers are highlighted in *blue color*, our response is provided in black color, and the text that was modified or added to the manuscript is marked with *red color*.

Response to the comments from Reviewer #1

Comment: *In the revised version of the manuscript by addressing the reviewers' comments, the authors have provided stronger claims in support of the uniqueness of their material, and I believe that this manuscript could be acceptable for publication after another round of revision.*

Response to the comment: We thank the Reviewer for their generally positive assessment of the work and the valuable suggestions. We appreciate that the Reviewer recognizes the relevance and the impact of the results presented in the manuscript. All suggestions from the Reviewer have been considered and addressed in the revised version of the manuscript.

Comment: *Concerning the controllability over the phase transition of molecular crystals, the authors are right in that “the phase transitions of molecular crystals are often hard to control once initiated, and in many cases the thermal hysteresis between the forward and reverse transition is significant. On the other hand, with respect to the shape transformation, some of the reported dynamic crystals show disintegration, splintering and deformation during the phase transition”. However, it should be acknowledged that the recent efforts in this direction have made some progress. By manipulating the phase boundaries, the thermal hysteresis of phase transition of the organic crystal can be greatly reduced to 2–3 °C, repeatable and reversible while retaining their crystalline nature (Small 2021, 17, 2006757). Moreover, a second-order phase transition can effectively decrease the rate of change in molecular configuration (J. Am. Chem. Soc. 1999, 121, 2808-2819), which might be also a direction towards the controllability over dynamic crystals. It would be greatly appreciated if these aspects can be discussed.*

Response to the comment: We thank the Reviewer for this comment, and for their valuable insight. We only recently became aware of one of these studies (Small, 2021). In the revised manuscript, we added a discussion on the recent efforts towards control of phase transitions in molecular crystals, and we also included the relevant references. To respond to the Reviewer's comment, the following text has been added to the manuscript with the relevant references:

“However, recent reports on altering the thermal hysteresis of phase transition by manipulating the phase boundaries⁴⁶ and the influence of second order phase transition on molecular configuration in a nickel complex have demonstrated the possible controllability of dynamic effects.⁴⁷”

46. Duan, Y., Semin, S., Tinnemans, P., Xu, J. & Rasing, T. Fully Controllable Structural Phase Transition in Thermomechanical Molecular Crystals with a Very Small Thermal Hysteresis. *Small*, 17, 2006757 (2021).

47. Falvello, L. R. *et al.* Tunable Molecular Distortion in a Nickel Complex Coupled to a Reversible Phase Transition in the Crystalline State. *J. Am. Chem. Soc.* 121, 2808–2819 (1999).

Response to the comments from Reviewer #2

Comment: *I am very satisfied that the authors have now added enough reported data to the supporting information for comparison with this work. The authors have addressed all comments, and the manuscript could be accepted in its present form.*

Response to the comment: We thank the Reviewer for their positive assessment of our work.

Response to the comments from Reviewer #3

Comment: *The authors have almost ignored all my comments and suggestions, and the current theoretical understanding is still very weak. I would suggest rejecting this paper.*

Response to the comment: We thank the Reviewer for their valuable time and feedback. We would like to reassure the Reviewer that in the first round of revisions we have considered carefully all of their comments and suggestions, and have done our best to address them all. In both the first and second round of revisions, the Reviewer has pointed that the theoretical understanding is “weak”, without providing specific constructive comments on the DFT-D model used or indeed, on the theoretical data presented. We appreciate that various computational approaches can be used to address—within the restrictions of time and computational cost—various aspects of the same problem. The Reviewer, who we feel is an expert in computational analysis of phase transitions, suggested to perform calculations for which we feel they are beyond the scope of our work presented here, and which would not, at least in our considerate view, change the conclusions already reached using the range of experimental and theoretical tools used. We would also like to bring to the attention of the Editor and Reviewer that we are unable to address some of the suggestions made concerning additional calculations as we lack the tools to perform these calculations at present.

Given all of the above, we would like to take this opportunity to explain better and in greater detail our response to each of the comments made by the Reviewer. Where possible and for the

sake of brevity, the original comments and our responses are grouped by topic in the text that follows.

1) The first set of comments made by the Reviewer was that the “*details of the calculations are not given anywhere in the manuscript or SI. We even don’t know which exchange function they are using and whether vdW is considered.*” We would like to bring to the Reviewer’s attention that both the methodology of the calculations and figures of the data produced from the calculations were provided as part of the Supporting Information during the initial submission of the manuscript. This included detailed methodological information on the exchange function(al) used, periodic geometry optimizations of the crystals, energetic barrier for cell transformations and simulations into the cooperative rotational energy barrier in GN. Moreover, this methodological information was accompanied by the results following these simulations and this data was also provided in the Supporting Information. Specifically, these details were available in the following section, which is copied here verbatim from the original submission:

“Equilibrium crystal structure of forms I and II of GN The experimental crystal structures for forms I and II of GN were subjected to periodic dispersion-corrected density functional theory (DFT-D) geometry optimization using CASTEP 8.02 as implemented in BIOVIA Materials Studio 8.03 using the PBE generalized gradient approximation (GGA) exchange–correlation functional⁴ and norm-conserving pseudopotentials.⁵ The D2 (G06) semi-empirical dispersion correction of Grimme⁶ was used. Brillouin zone integrations were performed on a symmetrized Monkhorst-Pack kpoint grid with a separation of 0.07 \AA^{-1} . The plane-wave basis set cut-off was set at 750 eV. The BFGS⁷ algorithm was used for the geometry optimization. The structural optimization was considered complete when the following convergence criteria were satisfied: maximum energy change of 1×10^{-5} eV per atom, maximum force of 3×10^{-2} eV \AA^{-1} , maximum stress of 5×10^{-2} GPa and maximum displacement of 1×10^{-3} \AA . The periodic DFT-D geometry optimization was performed first at fixed volume by optimizing only the atomic positions. This was followed by a second geometry optimization that relaxed both atomic positions and unit cell parameters. The resulting equilibrium structures for Forms I and II were used as the basis for further calculations as detailed below. Energetic barrier for cell transformation from form I to form II The CASTEP generated equilibrium crystal structure for Form I was used to generate a series of strained crystals. The strained crystals were generated by incrementally changing the cell lengths and β angles starting with the equilibrium cell geometry of form I and gradually transforming the cell parameters to those found in the CASTEP generated equilibrium crystal structure of Form II. A total of 6 crystals were generated: the strainfree form I equilibrium structure (strain step 0), 4 strained cells that represent the intermediate cell geometries during the martensitic phase transition and a sixth strained cell that has the equilibrium cell geometry found in the equilibrium form II structure. All 6 crystal structures were used as input for fixed-volume geometry optimization in VASP8-10 in order to estimate the strain energy in the GN crystal as it rapidly switches between forms I and II. VASP uses the projector-augmented wave (PAW)¹¹ method with planewave basis sets and PAW pseudo-potentials. In all calculations, the PBE⁴ generalized gradient approximation (GGA) was used for the exchange-correlational functional coupled with the D3 dispersion-correction.¹² VASP input files were generated using the cif2cell¹³ program. In all cases, a Γ -centered Monkhorst-Pack scheme was used to generate a tight K-point mesh with maximum K-point distance set to $2\pi \times 0.032 \text{ \AA}^{-1}$. In all calculations, the cut-off energy for the planewave basis was set to 700 eV. Within each self-consistent field

cycle, the convergence threshold was set at 1×10^{-4} eV and the geometry was considered converged when all forces were below 0.03 eV \AA^{-1} . The strain energy (ΔE) for each cell is reported relative to the equilibrium energy of the strain-free Form I crystal (strain step 0) as reported by VASP using the same convergence thresholds reported above. 3 Simulation of cooperative rotational energy barrier in GN In order to estimate the energy barrier for cooperative motion of the GN ions during the martensitic phase transition, a bilayer of hexagonal hydrogen-bonded sheets of GN ions were extracted from the equilibrium form I structure. All the ions in the top layer were rotated incrementally relative to the bottom layer at increments of 10° . The rotation angle is denoted ϕ . At each step, the energy of the bilayer ring topology was estimated using VASP. The VASP control parameters and convergence thresholds were the same as those outlined above. The rotational energy barrier (ΔE_{rot}) reported is relative to the energy of the ring topology found in the equilibrium form I structure ($\phi = 0$).”

In response to the Reviewer comments, in the first round of revision of the manuscript we moved the methodological information on the calculations to the main text so that all the methodological information could be found in the same location, making them more easily accessible to readers. After careful consideration and noting that the Reviewer did not object to the location of the theoretical figures, we did not feel it necessary to move the figures for the calculations to the main text as the calculations were performed with the aim to provide a support to the experimental work; we would like to keep the focus of this article on the experimental results, which are supported by computations. As such, the theoretical work is not the main focus of this manuscript.

“Equilibrium crystal structure of forms I and II of GN. The experimental crystal structures for forms I and II of GN were subjected to periodic dispersion-corrected density functional theory (DFT-D) geometry optimization using CASTEP 8.0⁶⁹ as implemented in BIOVIA Materials Studio 8.0⁷⁰ using the PBE generalized gradient approximation (GGA) exchange–correlation functional⁷¹ and norm-conserving pseudopotentials.⁷² The D2 (G06) semi-empirical dispersion correction of Grimme⁷³ was used. Brillouin zone integrations were performed on a symmetrized Monkhorst-Pack k -point grid with a separation of 0.07 \AA^{-1} . The plane-wave basis set cut-off was set at 750 eV. The BFGS⁷⁴ algorithm was used for the geometry optimization. The structural optimization was considered complete when the following convergence criteria were satisfied: maximum energy change of 1×10^{-5} eV per atom, maximum force of $3 \times 10^{-2} \text{ eV \AA}^{-1}$, maximum stress of 5×10^{-2} GPa and maximum displacement of $1 \times 10^{-3} \text{ \AA}$. The periodic DFT-D geometry optimization was performed first at fixed volume by optimizing only the atomic positions. This was followed by a second geometry optimization that relaxed both atomic positions and unit cell parameters. The resulting equilibrium structures for Forms I and II were used as the basis for further calculations as detailed below.

Energetic barrier for cell transformation from form I to form II. The CASTEP generated equilibrium crystal structure for Form I was used to generate a series of strained crystals. The strained crystals were generated by incrementally changing the cell lengths and β angles starting with the equilibrium cell geometry of form I and gradually transforming the cell parameters to those found in the CASTEP generated equilibrium crystal structure of Form II. A total of 6 crystals were generated: the strain-free form I equilibrium structure (strain step 0), 4 strained cells that represent the intermediate cell geometries during the martensitic phase transition and a sixth

strained cell that has the equilibrium cell geometry found in the equilibrium form II structure. All 6 crystal structures were used as input for fixed-volume geometry optimization in VASP⁷⁵⁻⁷⁷ in order to estimate the strain energy in the GN crystal as it rapidly switches between forms I and II. VASP uses the projector-augmented wave (PAW)⁷⁸ method with plane-wave basis sets and PAW pseudo-potentials. In all calculations, the PBE⁷¹ generalized gradient approximation (GGA) was used for the exchange-correlational functional coupled with the D3 dispersion-correction.⁷⁹ VASP input files were generated using the cif2cell⁸⁰ program. In all cases, a Γ -centered Monkhorst-Pack scheme was used to generate a tight K -point mesh with maximum K -point distance set to $2\pi \times 0.032 \text{ \AA}^{-1}$. In all calculations, the cut-off energy for the planewave basis was set to 700 eV. Within each self-consistent field cycle, the convergence threshold was set at 1×10^{-4} eV and the geometry was considered converged when all forces were below 0.03 eV \AA^{-1} . The strain energy (ΔE_s) for each cell is reported relative to the equilibrium energy of the strain-free Form I crystal (strain step 0) as reported by VASP using the same convergence thresholds reported above.

Simulation of cooperative rotational energy barrier in GN. In order to estimate the energy barrier for cooperative motion of the GN ions during the martensitic phase transition, a bilayer of hexagonal hydrogen-bonded sheets of GN ions were extracted from the equilibrium form I structure. All the ions in the top layer were rotated incrementally relative to the bottom layer at increments of 10° . The rotation angle is denoted ϕ . At each step, the energy of the bilayer ring topology was estimated using VASP. The VASP control parameters and convergence thresholds were the same as those outlined above. The rotational energy barrier (ΔE_{rot}) reported is relative to the energy of the ring topology found in the equilibrium form I structure ($\phi = 0^\circ$).

2) The second set of suggestions made by the Reviewer were that “*it would be better to do an NEB calculation to show the energy barrier for the forms I and II transition*” and that “*DFT calculations of the surface energies would give some insights*”. We appreciate that the nudged elastic band (NEB) theory is typically used to map reaction pathways or phase transitions when a careful understanding of the saddle points and energy barriers is important. In our work, however, the DFT-D calculations have confirmed that the two polymorphs are equienergetic in agreement with the rapid martensitic phase transition observed between forms I and II. We strongly feel therefore that NEB calculations would not change this conclusion and would only provide intermediate states during the phase transition, which are not central to this work. With regards to the suggestion that “DFT calculations of the surface energies would give some insights”, we have considered this suggestion carefully however the tools to accurately model the surface energies of these crystals using DFT methods are currently unaffordable to us.

In summary, we would like to thank once again the Reviewer for their time, effort and diligence in reviewing our manuscript on two occasions. We hope the above responses provide a clearer picture of our approach to the revisions and the decisions made concerning the above comments. A follow-up, more specialized research work that would focus on the computational aspects and benchmark different approaches in explaining various aspects of this phase transition could be an interesting continuation of the work presented in this manuscript, and we hope to be able to report such results in the future.